# Neuro-Cognitive Locomotion with Dynamic Attention on Topological Structure

Azhar Aulia Saputra [1], János Botzheim [2,*] and Naoyuki Kubota [1]

1    Graduate School of System Design, Tokyo Metropolitan University, 6-6 Asahigaoka,
     Hino 191-0065, Tokyo, Japan; aa.saputra@tmu.ac.jp (A.A.S.); kubota@tmu.ac.jp (N.K.)
2    Department of Artificial Intelligence, Faculty of Informatics, Eötvös Loránd University,
     Pázmány Péter sétány 1/A, H-1117 Budapest, Hungary
*    Correspondence: botzheim@inf.elte.hu

**Abstract:** This paper discusses a mechanism for integrating locomotion with cognition in robots. We demonstrate an attentional ability model that can dynamically change the focus of its perceptual area by integrating attention and perception to generate behavior. The proposed model considers both internal sensory information and also external sensory information. We also propose affordance detection that identifies different actions depending on the robot's immediate possibilities. Attention is represented in a topological structure generated by a growing neural gas that uses 3D point-cloud data. When the robot faces an obstacle, the topological map density increases in the suspected obstacle area. From here, affordance information is processed directly into the behavior pattern generator, which comprises interconnections between motor and internal sensory neurons. The attention model increases the density associated with the suspected obstacle to produce a detailed representation of the obstacle. Then, the robot processes the cognitive information to enact a short-term adaptation to its locomotion by changing its swing pattern or movement plan. To test the effectiveness of the proposed model, it is implemented in a computer simulation and also in a medium-sized, four-legged robot. The experiments validate the advantages in three categories: (1) Development of attention model using topological structure, (2) Integration between attention and affordance in moving behavior, (3) Integration of exteroceptive sensory information to lower-level control of locomotion generator.

**Keywords:** dynamic attention; topological map model; affordance detection; neuro-cognitive locomotion

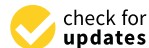



## 1. Introduction

When we talk about a legged locomotion system, one of the most important words that comes to mind is "dynamic". Nowadays, many researchers strive for dynamism in robot locomotion. An important question is, What should the indicators of dynamic locomotion be? To what extent can a robot's behavior imitate that of a natural animal?

The human locomotion system is not limited to generating motion. It also integrates several other aspects such as stability, perception, sensing, and memorization from the higher-level control systems, through the actuator parameter generator, to the movement behavior generator [1]. Researchers often impose constraints in order to simplify the integration of cognition and behavior generation, but this strategy may limit how much dynamic integration can be achieved. Even within such constraining simplifications, imitative models can be described from biological [2], mechanical [3–5], and cognitive viewpoints [6].

Animal (including human) movement will always involve sensorimotor coordination, i.e., an integration of sensory information with the actions realized by the motor neurons [7]. Sensorimotor coordination smoothly combines interoceptive and exteroceptive sensory information to produce a sense of orientation and movement. The visual system (exteroceptor) coordinates with the other sensory systems to influence the musculoskeletal system [8].

When a walking human faces a sudden obstacle during the leg-swinging phase (i.e., while the leg is lifted), the exteroceptive process plays a role in directly controlling the joints to avoid the obstacle. The visual information is processed in the cerebellum, basal ganglia, and supplementary motor areas. When sensory information from one system is inaccurate because of an injury, the central nervous system (CNS) will combine information from the other systems [9].

In dynamic locomotion, generated movement has to have an objective. According to Pfeifer, cognition, embodiment structure, and the locomotion generator should be integrated when developing a reliable neurobiological locomotion model [10]. By incorporating cognition into the model, it becomes possible to advance from sensory-response systems to a more sophisticated interpretative response system that can navigate dynamic environments. The difference is similar to the difference between a puppet and a living being: the puppet is always controlled, making no strategic decisions for itself, whereas a living being, on noticing a change in its surroundings, can change its movement strategy. Our goal is to realize a more human-like behavior by integrating neuro-cognitive functions into robot locomotion. The robot has to use both interoceptive sensory information about its own body and also exteroceptive sensory information about its surroundings. Integrating these sensory domains can be difficult when cognition is implemented separately from the locomotion generator. In current models of robot locomotion, most researchers consider only interoceptive information when building the locomotion generator. Some integrate locomotion and perceptional systems separately, implying that the perception information is not processed in low-level control in short-term adaptation.

From the current state of the art, we raised three scientific questions:

- How to build a locomotion model that integrates both exteroceptive and interoceptive sensory information?
- How can exteroceptive information affect lower-level locomotion control (short-term adaptation)?
- How can we build an attention model for robot locomotion using point-cloud data?

To solve these questions, we propose a neuro-cognitive locomotion model that integrates strongly with both exteroceptive and interoceptive sensory information. We develop a cognitive model useful within the lower-level locomotion model, which includes: (1) a dynamic attention model for exteroceptive sensory information, (2) a movement-related affordance detection model, (3) a locomotion generator model. The proposed model is implemented in simulated and real quadruped robot.

This paper is organized as follows. In Section 2 we discuss related research. In Section 3, we explain our proposed neuro-cognitive locomotion model. Then, Section 4 focuses on a perception model in short-term adaptation, explaining the Dynamic Attention model and Affordance detection model. After that, we describe the proposed locomotion generator, capable of considering external sensory input, in Section 5. In order to show the effectiveness of the proposed model, several experiments—both in computer simulation and in a real robot—are presented in Section 6. In Section 7, the advantage of the proposed technique and its comparison with existing model is discussed. Finally, in Section 8, discussion and conclusions are provided.

## 2. Current Research

Current research is surveyed in three categories: attention models in human and robots; perception and motion integration; and neural-based locomotion. We will address contributions in each category, towards developing a model for dynamically representing attention in topological structure. In general, the perceptual information is included only in higher-level locomotion controllers.

### 2.1. Attention Models in Humans and Robots

When walking freely through safe areas, we typically do not pay much attention to how we are moving, and sometimes not even to where we are. On encountering an obstacle

or disturbance, however, our attention focuses and intensifies to evaluate the obstacle, in order to form an appropriate response. In contrast, unsafe environments and uneven terrain keep our attention primed all the time, focusing intently on our surroundings and our bodies. We take care over every stepping action. Local perception, therefore, can change attention dynamically.

Gaze attention systems have been developed around image sensors [11,12]. These are inspired by neurobiological models and the behavior of the early primate visual system. Ude et al. implemented a gaze attention model in their humanoid robot [13]. The gaze concept has been implemented for mapping in robot soccer [14]. It has also been implemented for selecting an appropriate object in robot manipulation problems [15]. Nevertheless, image information requires extensive processing in order to represent an object's shape in 3D space. On the other hand, point-cloud data measured by a depth sensor can allow more efficient approaches to 3D representation [16,17]. In our proposed model, 3D information is essential to control leg swings. We use 3D point-cloud data generated by a depth sensor as visual input.

Processing the 3D point cloud requires a preliminary step to reduce computational costs. The points can be represented effectively and efficiently in a topological map model [18–20]. Current approaches to topological map representation need modification to work well with attentional systems, specifically to clarify the detailed surfaces of some objects. In this paper, we propose a dynamic-density topological map implemented in a growing neural gas network. The density of the network's nodes and edges dynamically adjusts to match attention areas. That adjustment inherently increases detail and clarity by increasing the information density where it is needed.

### 2.2. Perception and Motion Integration

Perceptual systems have been integrated in various ways with behavior. Vision sensors, for example, have been combined with a control system to detect obstacles and generate context-appropriate robot behavior [21]. However, most of the locomotion and perceptional systems were developed separately: a self-contained perception mechanism handles obstacle avoidance and its output—a movement plan—is used as input for the locomotion mechanism. This approach has been successfully implemented in a legged robot. Barron et al. [22] implemented perception-based locomotion in a hexapod robot. They used visual information from the perception mechanism in a central pattern generator (CPG) to process obstacle avoidance and motion guidance. Manchester et al. [23] installed more complex external sensors to reconstruct a ground surface and generate the sequences of foothold position. The system's high computational overhead cannot respond to short-term changes and involves footstep planning at every footstep. There are also many other footstep planning strategies implemented in biped robots [16,24,25] and hexapod robots [26]. On the other hand, other researchers used a closed-loop model to integrate perception and action to relate robot embodiment with environmental conditions. The appropriate gait or locomotion rate is adjusted in response to detected environmental changes [27]. This approach reconstructs the map before planning the motion and controls only high-level motion command, such as moving speed. Nevertheless, the stability model of legged robot comprises low-level motion control. Several approaches explain how external sensory information plays a role in low-level control. Therefore, the proposed model includes a role for cognition in lower-level control of locomotion. We choose a time-of-flight sensor as an external sensor which can detect object shapes at a lower cost than similarly capable vision sensors.

Human movement does not require conscious planning of every detail; it may be largely generated on the fly as a consequence of elementary behaviors underpinning steering and obstacle avoidance [28]. In current systems, perceptual information is used to control higher-level motion planning, such as path planning or walking-plan generation. Therefore, we propose a neural-based locomotion that can deal with both high-level planning as well as short-term adaptation, processing perceptual information in both regimes. It can respond to upcoming obstacles that appear suddenly during the limb-swinging process.

### 2.3. Neural-Based Locomotion Generator

Another way to achieve dynamic locomotion is the neural-based approach [29–32], which has found extensive application in quadruped robot locomotion [3,33–35]. Kimura et al. developed a neural model and its reflex mechanism for generating motion patterns, effective in a quadruped robot on rough terrain [34]. Maufroy et al. also developed a neural model for generating a motion pattern that changes dynamically depending on the desired speed [36]. Neural systems have also been developed that transition robots from one motion pattern to another [37], for example, adjusting gait for greater energy-efficiency at different speeds [35]. On the other hand, a neural-based control model has been used for controlling an exoskeleton actuator [32]. In our previous research, we proposed a neural-based locomotion planner for both higher-level and medium-level controllers [38–40], using body posture and contact points as feedback signals.

However, the current state of the neural-based locomotion model does not consider exteroceptive sensory information, focusing more on interoceptive information. Therefore, using the current state model, the robot can not give a response if there is an upcoming obstacle. Full dynamic locomotion, however, should include the ability to avoid obstacles before contact, including obstacles that appear without warning, so the motion controller must be able to process external perceptual information for short adaptations. In this paper, we propose a neural-based locomotion system that uses perceptual information detected by a dynamic attention system for all of short-, medium-, and long-term adaptation.

### 3. System Overview

The overall system model can be seen in Figure 1. It represents a neuro-cognitive locomotion system which integrates internal and external sensory information. The system is segmented into short-, medium-, and long-term adaptation. Different terms of adaptation represent the size of data. Short-term adaptation needs only a single sensor data for direct perception, medium-term adaptation needs a batch of sensor data for recognition, and long-term adaptation needs larger sensor data for environmental building. It also represents the coverage action: short-term adaptation has a small action changing in joint angle level every time cycle, medium-term adaptation has larger action in each step length, and long-term adaptation has the largest action area which composes a bench of movement provision. It also represents the time action: short-term adaptation is updated in every time cycle, medium-term adaptation in every step length, and long-term adaptation if there are different intentions.

In this paper, we focus on a short-term adaptation system as our novel contribution. Short-term adaptation requires a direct response to detected changes in every time cycle. To achieve this, we use point-cloud data from the external sensors. First, to reduce data representation overheads, we use the dynamic attention model explained in Section 4.1 to generate a topological map model in a neural gas network with a dynamic node density. The network's node density represents the degree of attention to corresponding regions. The Dynamic Attention model outputs a $3 \times A$ matrix to represent the network's nodes, and an $A \times A$ matrix to represent its edges, where $A$ is the number of nodes.

After that, the dynamic attention matrices are interpreted by an Affordance Detection (see Section 4.2) system. Detected objects are represented by a group of triangular planes whose size represents the degree of stepping precision. The Affordance Detection system then outputs the size of each area, and an associated normal vector representing how safe that area is. If there is a nonhomogeneous normal vector for any area, the Dynamic Attention system increases the density of nodes there. For example, when the robot faces a suspected obstacle, the topological structure in the suspected area becomes denser so that more information will be collected for analysis. The Affordance result is processed also to detect objects for medium-term adaptation. Affordance results will be considered by the path-planning system in the higher-level controller. Depending on the intention, motivation, and goal position given by a human operator, the path planner sets an overall speed and direction.

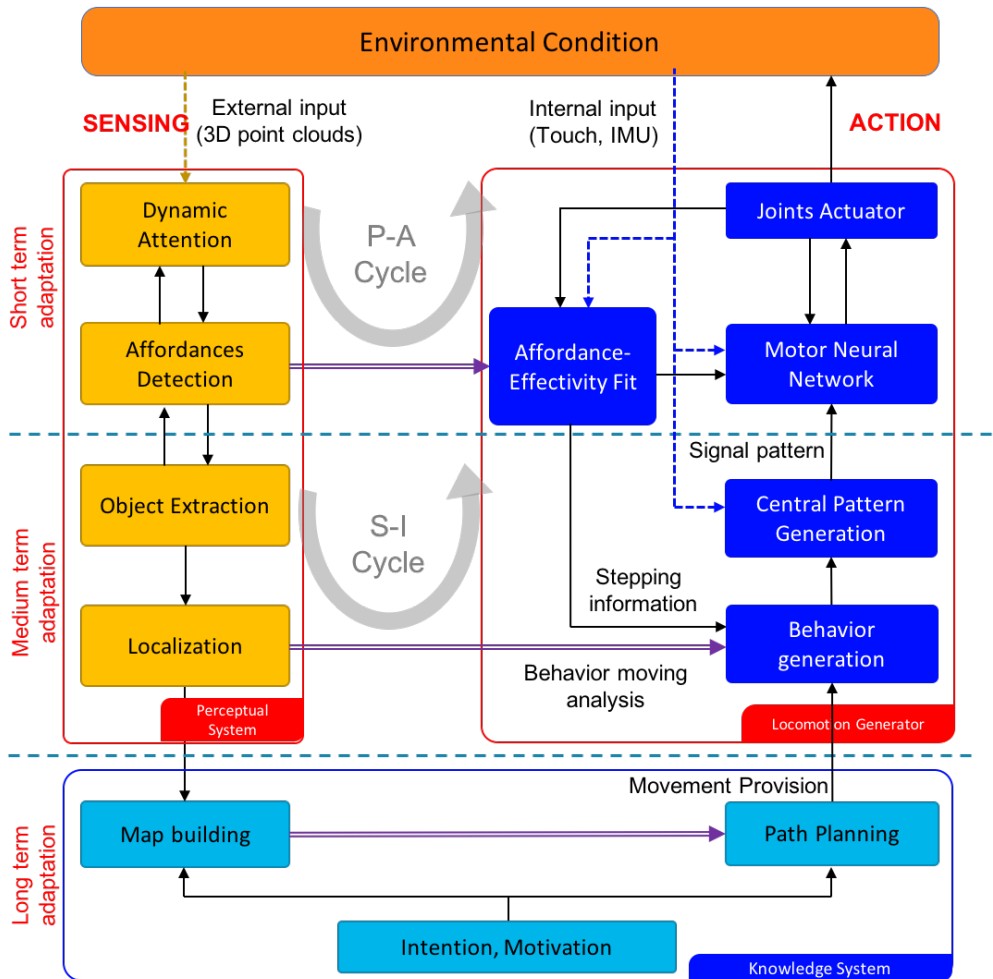

**Figure 1.** Diagram of neuro-cognitive locomotion.

In order to generate appropriate action and integrate the Affordance Detector with the locomotion generator, we build an *Affordance Effectivity Fit* (AEF) process. The AEF determines whether an object affects the robot's immediate needs: when the object is close enough, the AEF will interrupt and control the motor neural network (lower level), the degree of interruption, and the swing direction. Swing generation, from a coupled neural oscillator, controls the angular velocities of joints in each leg. The stepping precision will be affected by the intention of the controller for positioning the feet in the target area. In some conditions, the AEF process will change the walking provision to encourage medium-term adaptation. Details will be explained in Section 5. The flow of figure description from overall system design to locomotion generator structure to the *Affordance Effectivity fit* model can be seen in Figure 2.

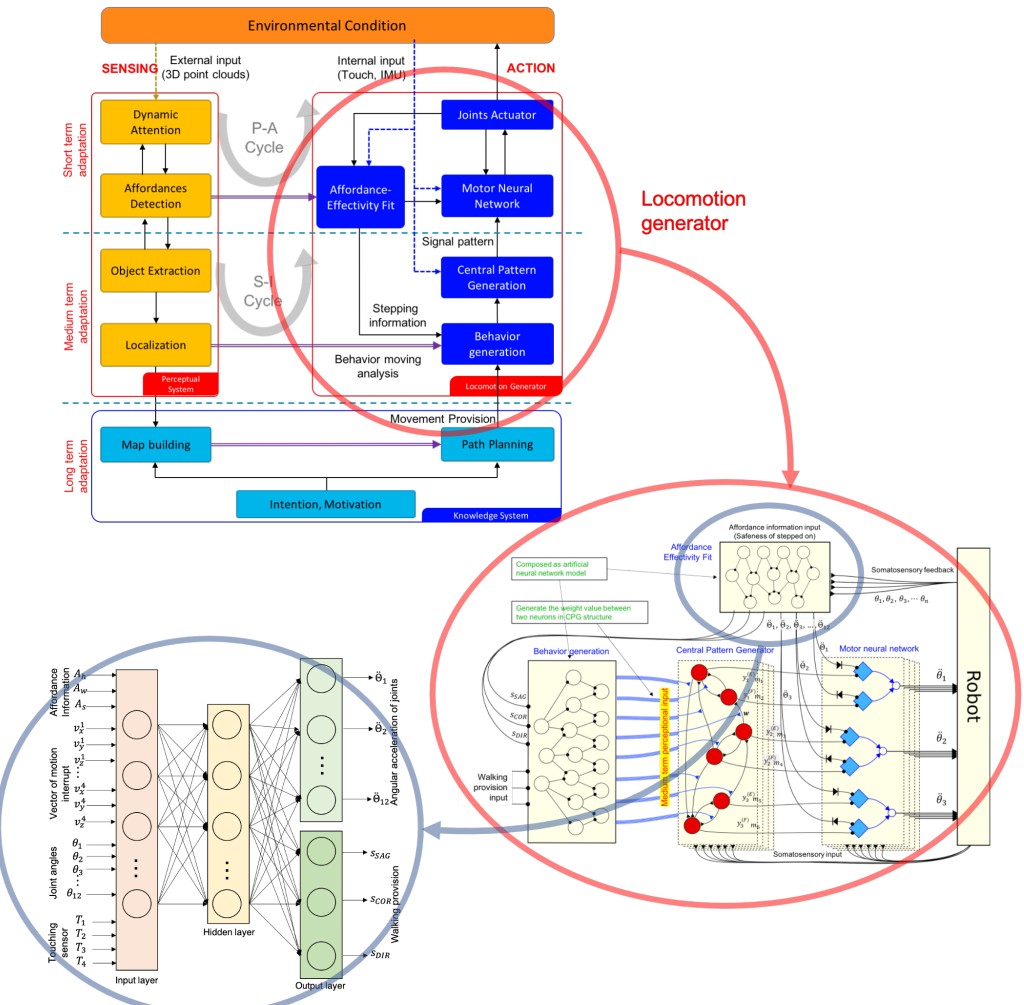

**Figure 2.** Flow of figure description from overall system design to locomotion generator structure to *Affordance Effectivity fit* model.

## 4. Perception Model

This section emphasizes the perception model in short-term adaptation, explaining the Dynamic Attention model and Affordance Detection model.

### 4.1. Dynamic Attention with Density Representation

Attention can often represent an animal's movement intentions. When intending to go straight ahead, the animal focuses in front, ready to react to suspected obstacles. On detecting an obstacle, attention is redirected to analyze the suspected obstacle in detail. If the obstacle is deemed harmless, the attention is moved away [41].

The proposed dynamic attention model realized the mechanism of attention using growing neural gas. The gas density manifests in the number and distribution of nodes and edges throughout the space. The attention process entails controlling the density of data representation to focus on specific attention areas, being circles of a particular radius, centered on a suspected node. We use four Pico Flexx time-of-flight sensors, developed by PMD technologies [42]. Each sensor captures depth information only, covering a $45° \times 62°$ field of view. The four sensors are arranged as depicted in Figure 3 for a composite view covering $165° \times 62°$. The depth information that they gather is processed into a topological structure model using unsupervised learning. The basic Growing Neural Gas (GNG) technique can be implemented for real-time topological map generation as proposed by Toda et al. [20].

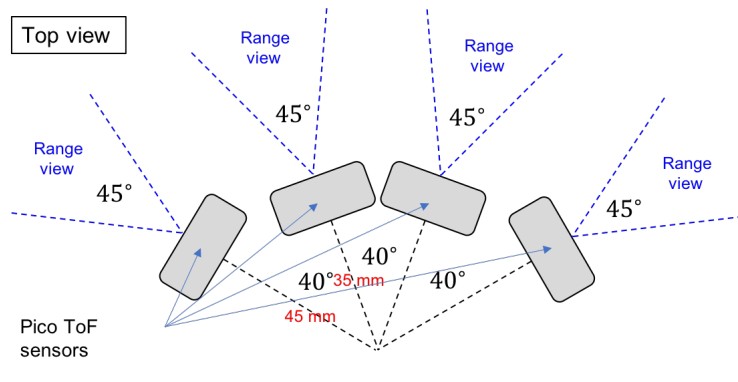

(**a**)

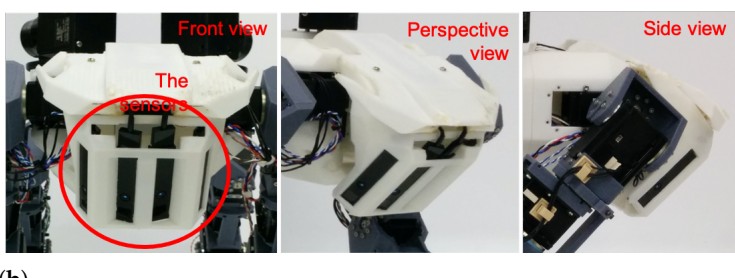

(**b**)

**Figure 3.** Design of time-of-flight sensors. (**a**) Structure of sensor placement. (**b**) Appearance of time of flight sensors from three different viewpoints: front, perspective, and side view.

The topology-represented dynamic attentions process is grounded in the Dynamic Density Growing Neural Gas (DD-GNG) [43], shown in Algorithm 1. In our system, the raw data ($\mathbb{P}$) around the suspected obstacle are processed separately.

---

**Algorithm 1** Dynamic Topological Structure

---

1: **Init:** generate two nodes at random position
2:     $c_{i,j} = 0 \ (\forall_i, \forall_j)$, $A = 1, 2$, $r = 2, t = 0$
3: **loop**
4:     $\mathbb{P} \leftarrow$ raw data of proposed sensors
5:     $\lambda \leftarrow 0$
6:     **for** $i \leftarrow 1$ to $\max_\lambda$ **do**
7:        **if** $t \mod 2 = 0 \wedge$ obstacle detected **then**
8:           $V_\lambda \leftarrow$ a random of $\mathbb{P}$ in suspected obstacle
9:        **else**
10:          $V_\lambda \leftarrow$ a random of $\mathbb{P}$
11:        $\lambda \leftarrow \lambda + 1$
12:     GNG main process ($V$) // (main layer)
13:     **if** $r > \gamma_r$ **then**
14:        Triangulation process ($\mathbf{c}, \mathbf{h}$)
15:        $\mathbf{L} \leftarrow$ Segmentation Process ($\mathbf{c}, \mathbf{h}$)
16:        $\mathbf{L}_{suspected} \leftarrow$ Suspected obstacle detection ($\mathbf{L}$)
17:        $\mathbf{L}_{suspected}^{(size)} \leftarrow$ Get suspected obstacle area ($\mathbf{L}$)

---

The initialization process begins by generating two initial random nodes. After that, in the main loop, $V$ is generated as raw point-cloud input data. Then, the main GNG process is performed as presented in Algorithm 2. The parameter notation can be seen in Table 1.

In the proposed model, we add a parameter representing the strength of node ($\delta$) calculated by considering suspected node. It is calculated when the node is created. If the

node has minimum utility value and $E_u / U_l > k \cdot \delta_u$, then the node will be removed. Parameter $E_u$ is the local error variable of node $u$, $U_l$ is the utility value of node $l$, $\delta_u$ is the degree of strength of node $u$, and $k$ is the parameter for representing degree of deletion. We perform triangulation and segmentation if the number of nodes is larger than the threshold ($r > \gamma_r$). The algorithm of suspected node can be seen in Algorithm 3.

---

**Algorithm 2** DD-GNG main process

---

1: **for** $t \leftarrow 1$ to $\lambda$ **do**
2:      Generate random input data $v_t$
3:      $s_1 = \arg\min_{i \in A} ||w \cdot (v_t - h_i)||$
4:      $s_2 = \arg\min_{i \in A \setminus s_1} ||w \cdot (v_t - h_i)||$
5:      **if** $C_{s_1,s_2} = 0$ **then**
6:          $C_{s_1,s_2} = 1$
7:      $g_{s_1,s_2} = 0$
8:      $E_{s_1} \leftarrow E_{s_1} + ||w \cdot (v_t - h_i)||$
9:      $U_{s_1} \leftarrow U_{s_1} + ||w \cdot (v_t - h_2)|| - ||w \cdot (v_t - h_i)||$
10:      $h_{s_1} \leftarrow h_{s_1} + \eta_1 \cdot (v_t - h_i)$
11:      **if** $C_{s_1,j} = 0$ **then**
12:          $h_j \leftarrow h_j + \eta_2 \cdot (v_t - h_j)$
13:          $g_{s_1,j} \leftarrow g_{s_1,j} + 1$
14:          **if** $g_{s_1,j} < g_{max}$ **then**
15:              $C_{s_1,j} = 0$
16:      **if** $(t \equiv \kappa) = 1$ **then**
17:          $u = \arg\max A_{i \in A} E_i$
18:          $l = \arg\min A_{i \in A} U_i$
19:          **if** $E_u / U_l < k \cdot \delta_u$ **then**
20:              $A \leftarrow A - l$
21:      $E_i \leftarrow E_i - \beta E_i (\forall i)$
22:      $U_i \leftarrow U_i - \chi U_i (\forall i)$
23: **if** $A <$ Max. Num. of Nodes **then**
24:      $r \leftarrow r + 1$
25:      $u = \arg\max A_{i \in A} E_i$, $f = \arg\max A_{C_{u,i}=0} E_i$
26:      $h_r = 0.5(h_u + h_f)$
27:      $E_u \leftarrow E_u - \alpha E_u$, $E_f \leftarrow E_f - \alpha E_f$
28:      $E_r = 0.5(E_u + E_f)$
29:      $\delta_r =$ generate the strength of node ($h_r$)
30:      $A \leftarrow A + r$

---

**Algorithm 3** Generate the strength of node ($h_i$)

---

1: $\delta \leftarrow 1$
2: **for** $j \leftarrow 1$ to $Ob^{(num)}$ **do**
3:      **if** $Ob_j$ is suspected obstacle **then**
4:          $A_{min} = Ob_i^{(position)} - Ob_i^{(size)} / 2$
5:          $A_{max} = Ob_i^{(position)} + Ob_i^{(size)} / 2$
6:          **if** $h_i > A_{min} \wedge h_i < A_{max}$ **then**
7:              $\delta \leftarrow \delta + 1$
8: **return** $\delta_i$

**Table 1.** DD-GNG Parameters Notation.

| | |
|---|---|
| $h_i$ | the 3D position of $i$th node |
| $w$ | the weight value of node |
| $A$ | set of 3D node positions |
| $C_{ij}$ | edge values between the $i$-th and $j$-th nodes |
| $g_{ij}$ | age values between the $i$-th and $j$-th nodes |
| $s_1$ | first nearest nodes |
| $s_2$ | second nearest nodes |

### 4.2. Affordance Detection

Affordance is what the environment offers to individuals introduced by Gibson [44]. Affordance does not depend on the ability of the individual to recognize or use it. Affordance is also defined by Turvey as the dispositional properties of the environment [45], complemented by the effectivity or dispositional properties of the actor. Affordance is hence not inherent to the environment; it depends also on particular actions and the actor's capabilities. Differences between in individuals' bodies may lead to different perceptions of affordance. Perception also depends on the intention; while affordances exist in the environment regardless of whether they are perceived, perception of an affordance depends on the affordance being there to create it.

Animal locomotion is controlled by perceiving affordances [46]. In other words, prospective action is generated depending on the affordance information that the locomotion generator receives [47]. In free space, animal stepping movements are governed according to the body's inertial condition. The adaptation process compares the estimated next stepping point, accounting for current inertial conditions, with the affordances of the surface.

In our model, affordance is represented using a collection of triangular facets and normal vectors, approximating the physical surface. The illustration of the proposed affordance can be seen in Figure 4. It received topological information composed as nodes ($h_{ij}$) and their edges ($C_{ij}$) from the Dynamic Attention model explained in Section 4.1. Each facet has a degree of stepping stability calculated from the degree of safeness. The first step in developing this representation is to triangulate the surface explained in the previous research [43] After triangulation, each triangle's normal vector is calculated by Equation (1), as illustrated in Figure 5. $\mathbf{n}_0$, $\mathbf{n}_1$, and $\mathbf{n}_2$ are the three DD-GNG nodes ($\mathbf{h}$) at the triangle's vertices.

$$\mathbf{N_i} = \frac{(\mathbf{n}_0 - \mathbf{n}_1) \times (\mathbf{n}_0 - \mathbf{n}_2)}{||(\mathbf{n}_0 - \mathbf{n}_1) \times (\mathbf{n}_0 - \mathbf{n}_2)||} \tag{1}$$

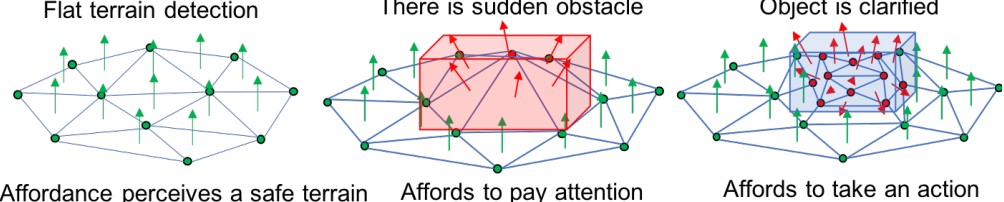

Flat terrain detection     There is sudden obstacle     Object is clarified

Affordance perceives a safe terrain    Affords to pay attention    Affords to take an action

**Figure 4.** Affordances of terrain and integration with attention model.

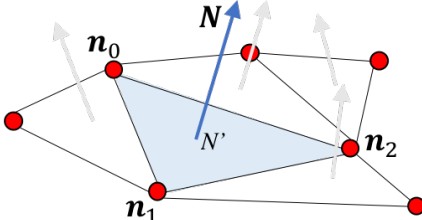

**Figure 5.** Calculation of normal vector.

After calculating the normal vector for every triangular facet, we calculate the weight connection between one facet and the surrounding facets in Equation (2); each facet thus has up to three connections.

$$w_{ij} = \cos^{-1}\left(\frac{\overrightarrow{\mathbf{N}}_i \cdot \overrightarrow{\mathbf{N}}_j}{||\overrightarrow{\mathbf{N}}_i|| \cdot ||\overrightarrow{\mathbf{N}}_j||}\right) \tag{2}$$

4.2.1. Object Segmentation Process

After building the weight connections, we segment the facet collection into broader surfaces corresponding to walls, safe terrain, and unknown objects. The segmentation process can be seen in our previous model [43].

In the segmentation process, we collect the number of labels ($n$), the number of facets in each label ($L$), and the normal vectors of the facets in every label ($\overrightarrow{\mathbf{T}}$) calculated by $\overrightarrow{Ob^{(v)}}_i = \frac{1}{L_i}\sum_{j=0}^{L_i} \overrightarrow{\mathbf{T}}_{i,j}$. The segmentation process generates both the object position ($Ob^{(\text{pos})} = \frac{1}{L_i}\sum_{j=0}^{L_i} \overrightarrow{\mathbf{H}}_{i,j}$) and also the object size ($Ob^{(\text{size})}$). This information will also become the affordance information input of the AEF structure in Section 5.2, where the object's height is calculated as $A_h = \sum_{j=1}^{L_i} \arg max(H_{i,j}^z)$, object's width is calculated as $A_w = \arg max||Ob^{(\text{pos})} - H_{i,j}^y||, j \in L_i$, and object's length is calculated as $A_s = \arg max||Ob^{(\text{pos})} - H_{i,j}^x||, j \in L_i$, where, $L_i$ is the number of facets in the $i$th label, $H_{i,j}$ is the center position of the $j$th facet in the $i$th object label.

Before the classification, the orientation of every surrounding surface needs to be calculated by Equation (3), from which safeness is calculated by comparison with the vertical unit vector.

$$\gamma_i = \cos^{-1}\left(\frac{\overrightarrow{Ob^{(v)}}_i \cdot \overrightarrow{\mathbf{N}}_{(ver)}}{||\overrightarrow{Ob^{(v)}}_i|| \cdot ||\overrightarrow{\mathbf{N}}_{(ver)}||}\right), \mathbf{N}_{(ver)} = \{0, 1, 0\} \tag{3}$$

Whenever the density of perceived affordance $d$, calculated by Equation (4), is lower than the threshold ($v$), then it will show the integration between affordance and attention. If $d > v$, then the perception will change the robot's behavior.

$$d = \text{Number of nodes in object/Object size} \tag{4}$$

4.2.2. Integration with Attention

Awareness and Attention influence the perception result of Affordance. The Affordance can be misperceived when there is low accuracy in attention and awareness [48]. When we are walking in a lonely road, our attention may not focus on the way. In this case, the awareness is low. When there is an obstacle coming, then the focus is suddenly directed to the obstacle [49]. The human process is efficient; the attention is automatically changed when it is required.

In our model, terrain roughness is analyzed from the coherency of the normal vectors of the triangular mesh facets ($\mathbf{N_i}$). Depending on whether the normal vectors in the robot's

path deviate from the vertical, the affordance process increases the strength values of associated nodes ($\delta_i$).

### 4.2.3. Integration with Effectivity

For actual usefulness, affordance needs to be integrated with effectivity. Effectivity provides a way to consider whether the individual robot or animal can actually make use of the affordance. Affordance and effectivity are hence important, and they form the ecological boundary of the perceiving–acting cycle that allows actions to succeed in attaining their goals [50]. In our locomotion system, the effectivity of behavior complements the affordance information. In short-term adaptation control, two effectivities are used: the prediction of the next stepping point, and the inertial leg movement during the swing. The affordance perceiver will send the possible swing action to the behavior interrupt. This information will then be considered for generating the appropriate action, whether as short-term behavior or in medium-term adaptation (see Figure 1). In our proposed model, the Affordance–Effectivity Fit is part of the locomotion generator explained in Section 5.

## 5. Locomotion Generator

The locomotion generator is a part of the overall system. The structure of the locomotion model can be seen in Figure 6. The locomotion generator is processed in short-term (every timestep) and medium-term (every moving step) adaptation. CPG is processed in medium-term adaptation while the MN and AEF module are processed in short-term adaptation.

The MN module received somatosensory input and motion pattern input from the CPG and receives interrupt commands (joint signals) from the AEF process. Furthermore, the AEF model is composed as the artificial neural network (ANN) model to generate correction signals to motor neurons based on the affordance information and the robot's current condition. From the combination of the external sensory information (affordance perception) and internal sensory information (robot's current condition), the dynamic locomotion is generated.

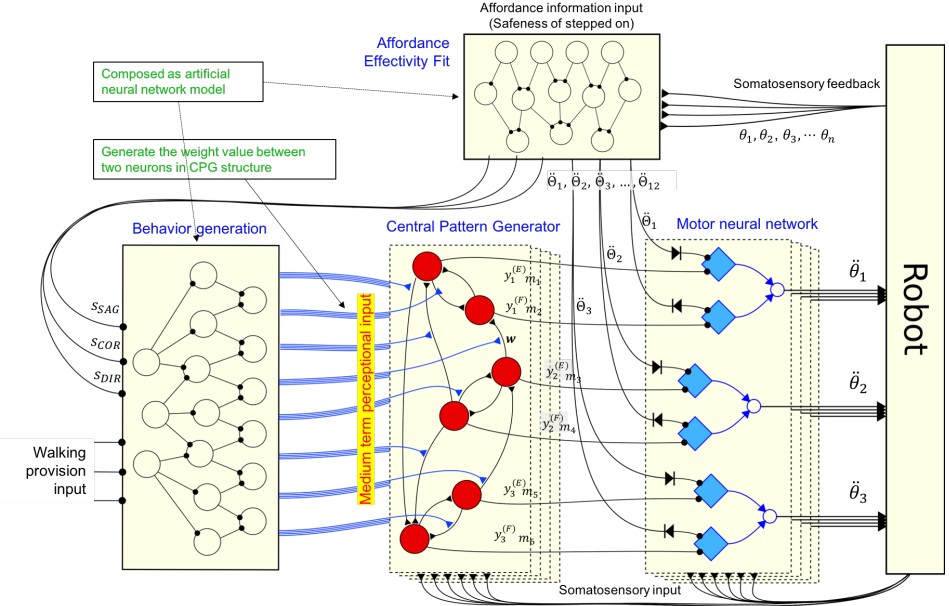

**Figure 6.** Neural interconnection model of the proposed locomotion generator.

### 5.1. Central Pattern Generation

The Central Pattern Generator controls the motion pattern over medium-term adaptation. CPG produces dynamic rhythmic signals by exploiting mutual inhibition of coupled neurons. We used the neural oscillator proposed by Matsuoka [51]. The rhythmic signal generated in our locomotion model is influenced by the CPG structure generated by behav-

ior generation (see Figure 1) and also by feedback from sensory information. One coupled CPG neuron represents one actuator or one joint angle. Since there are 12 active joints in our four-legged robot, each incorporating one actuator, 24 CPG neurons are required. The mathematical model can be seen in Equations (5)–(7).

$$\tau \frac{\mathrm{d}}{\mathrm{d}t} x_i = \left( x_0 - x_i - \sum_{j=1}^{n} w_{ij} y_j + \sum_{l=1}^{n} V_{il} s_l - b v_i \right) \tau_f \tag{5}$$

$$T \frac{\mathrm{d}}{\mathrm{d}t} v_i = (y_i - v_i) \tau_f \tag{6}$$

$$y_i = \max(x_i, 0), \tag{7}$$

where $x_i$ and $y_i$ represent the $i$th neuron's inner state and output signal, respectively and $v_i$ is the neuron's self-inhibition effect. Connections $w_{ij}$ are between neurons $i$ and $j$; connections $V_{il}$ are between neuron $i$ and the $l$th sensory input, $s_l$. To implement adaptation, $b$ is required. The time constant of the inner state and the adaptation effect are represented by parameters $\tau$ and $\tau_f$. The inner state value ($y_i$) will be transferred to the MN structure, influenced by the synaptic weight between $i$th of CPG neuron and the $i$th MN neuron ($m_i$).

### 5.2. Affordance–Effectivity Fit

The ANN process integrates the affordance perception and the robot's effectivity to generate appropriate action. This novel approach can interrupt the motion pattern to avoid an immediate obstacle or control the walking gait. The model applies perceptual information generated by the affordance detection model (as described in Section 4.2).

In human and animal movement, it is still not clear how affordance and effectivity are represented in the brain, and more specifically in the sensorimotor cortex (SMC) [52]. However, kinematic parameters and kinetic parameters can represent the human and animal movement [53]. The movement of the human hand involves a synergy of muscle contraction, velocity, and acceleration, automatically. One of the most prominent human movement models is Optimal Feedback Control [54]. In this model, the primary motor cortex sends top-down motor commands. By using feedback signals from primary somatosensory cortex, instantaneous error can be determined by comparing the motor command with an efferent copy of the motor command created by an optimal feedback law [55].

In our model, we used both kinematic and kinetic parameters as input, and used the posture and movement generated from somatosensory cortex as feedback. Since the joints are built around angle-based actuators, the sensors measure angular velocity, direction of motion, and the joints' angular displacements. From all this information, the processor generates the angular velocity of joints and moving gain as its output. Our model is implemented in an artificial neural network in order to decrease the computational complexity.

The artificial AEF neural network is depicted in Figure 7. There are 31 neurons representing 31 parameters in the input layer. Three parameters represent the output of the affordance detector (size of unsafe area ($A_h$, $A_w$, and $A_s$)); four 3D vectors ($v^l_{x,y,z}$) represent 3D interrupt direction of the leg's swinging movement; 12 inputs for angle of joint ($\theta_1, \theta_2, \theta_3, \cdots, \theta_{12}$); four inputs represent the ground force sensors ($T_1, T_2, T_3, T_4$) attached in the soles of the feet.

There are two groups of output neurons that are activated alternately. The first group of output layers is composed of 12 angular acceleration of joints ($\ddot{\Theta}_1, \ddot{\Theta}_2, \ddot{\Theta}_3, \cdots, \ddot{\Theta}_{12}$). It is generated if there is an interrupt signal from the Affordance–Effectivity module to the MN module. Second, there are three parameters ($s_{SAG}, s_{COR}, s_{DIR}$), representing the walking provision composed as sagittal speed, coronal speed, and direction. This input will be transferred to the CPG module.

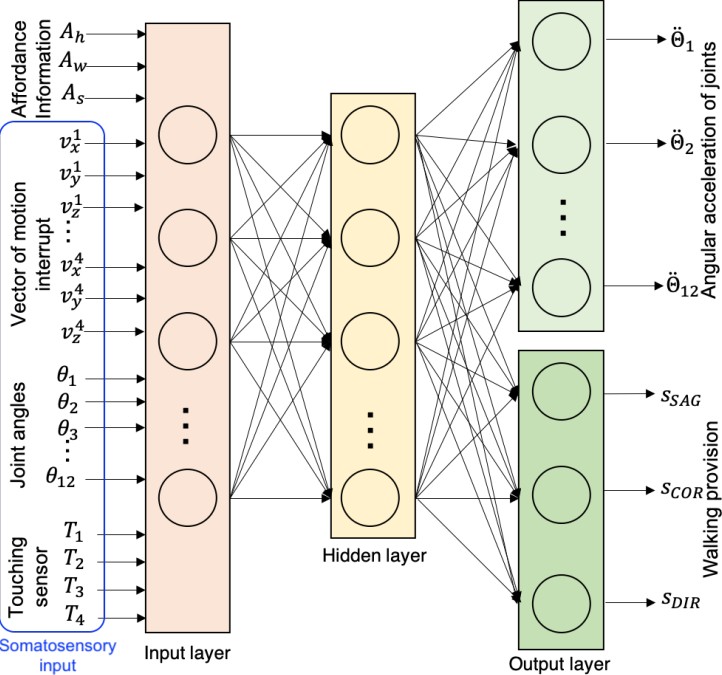

**Figure 7.** ANN model for Affordance–Effectivity Fit.

*5.3. Motor Neuron Pools*

The motor neurons control the pattern neuron for certain joint angles. The MN process receives input from the CPG and AEF processes. Each joint is composed of two motor neurons. It represents the mechanism of the flexor and extensor muscles to move the joint. To match the joint's angle-based mechanical structure, the MN module produces the angular acceleration of joints.

$$\ddot{\theta}_i = (\mu_1 \ddot{y}_i + \mu_2 \Theta_j) \cdot \ddot{\theta}_{MAX} \cdot F_l(\theta_i) \cdot F_v(\dot{\theta}_i) + S \tag{8}$$

In Equation (8), signal outputs from the central pattern generator ($\ddot{y}_i$) and motion interrupt generator ($\Theta_i$) are used as the signal activation of the MN process. $\mu_1$ and $\mu_2$ represent the gain parameters of the input signal. Every joint has its own maximum acceleration $\ddot{\theta}_{MAX}$ limit, chosen through preliminary testing. Parameter ($F_l(\theta_i)$) represents force–length value, parameter ($F_v(\dot{\theta}_i)$) represents force–velocity value, and parameter ($S$) represents internal sensory input. $\theta_i$ and $\dot{\theta}_i$ are the current angle value and angular velocity of the $i$th joint.

The force–length value and force—velocity value are calculated in Equations (9) and (10), following the hill-type equation model [56].

$$F_l(\theta_i) = e^{\left( c \left| \frac{\theta_i - \theta_i^{opt}}{\theta_R \omega} \right|^3 \right)} \tag{9}$$

$$F_v(\dot{\theta}_i) = \begin{cases} \frac{\dot{\theta}_{MAX} - \dot{\theta}_i}{\dot{\theta}_{MAX} - \dot{\theta}_i \cdot K} & \text{if } \dot{\theta}_i < 0 \\ N + (N+1)\frac{\dot{\theta}_{MAX} - \dot{\theta}_i}{7.56 K \dot{\theta}_i - \dot{\theta}_{MAX}} & \text{if } \dot{\theta}_i \geq 0 \end{cases} \tag{10}$$

In Equation (9), parameter $\theta_i^{opt}$ represents optimal value of joint angle for maximal generated output force. Parameter $c$ is computed as $ln(0.05)$, where $F_l(\theta^{opt}(1 \pm \omega)) = 0.05$.

In Equation (10), ($\dot{\theta}_i < 0$) indicates a muscle-shortening condition. Parameters $\dot{\theta}_{MAX}$ and $K$ represent the maximum velocity contraction velocity and constant value. Then, $\dot{\theta}_i \geq 0$ indicates muscle lengthening condition. Parameter $N$ represents the dimensionless amount of force $\ddot{\theta}/\ddot{\theta}_{MAX}$. It is acquired at a lengthening velocity $\dot{\theta} = -\dot{\theta}_{MAX}$.

Furthermore, the input calculation from the internal sensory information is as Equation (11)

$$S = \sum_{l=1}^{n} z_{il} s_l \tag{11}$$

where $n$ is the number of sensory interconnections. Parameter ($z_{il}$) represents the weight connection between the $i$th MN neuron and the $l$th sensory input ($s_l$). Sensory input comes from the four ground-touching sensors; one of each is in each leg. The MN output is the value of joint angular acceleration calculated in Equation (8), transferred to the robot actuators.

## 6. Experimental Results

In this section, we will show several experiments to confirm the effectiveness of each of the modules listed below:

1.  To show the effectiveness of the attention model using 3D point-cloud data, the proposed dynamic attention model will be tested and compared with the current model in a computer simulation.
2.  To analyze the effectiveness of the proposed affordance detection and its integration with the attention model, we will show the result of the proposed affordance detector with the dynamic topological map as its input. Then, we show the integration between perceived affordance information and attention.
3.  To test the effectiveness of the locomotion generator, we optimize the interconnection structure of the central pattern generation and motor neuron pools.
4.  To evaluate the effect of the robot's posture on its action generation, we analyze the affordance–effectivity fit model. The ANN of affordance–effectivity fit should be optimized beforehand.
5.  Finally, in order to prove the effectiveness of the whole proposed model, we realize the computational model in ODE and a real four-legged robot explained in [57] and conduct quantitative experiments. The robot's structure can be seen in Figure 8.

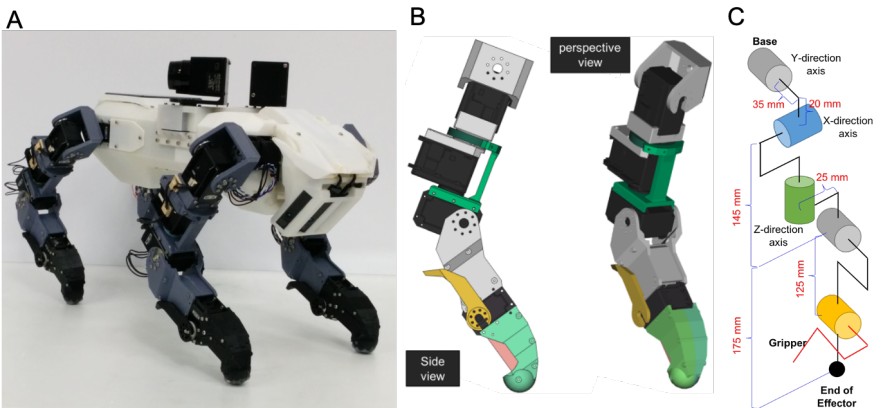

**Figure 8.** Design of the robot in the simulation. (**A**) shows the visualisation of the robot. (**B**) shows the CAD design of the robot's leg. (**C**) shows the joint structure of robot's leg.

### 6.1. Dynamic Attention Experiment

This experiment shows how the topological map can represent the attention model for 3D point-cloud data. We show that the density of the topological node increases in the appropriate area. The process takes depth sensory information as input data. The attentional information is then represented by the dynamic density of the topological map. The parameters are set as in Table 2. The parameter setting was acquired from the preliminary test in our previous studies [43]. However, to set the parameter $\lambda$, we have to consider the number of raw point cloud data, controller performance, and desired time step. In our system, we have 154112 raw point-cloud data. Therefore, we set $\lambda$ around 0.2 percent of the total point-cloud data.

**Table 2.** DD-GNG Parameters.

| $\lambda$ | $\kappa$ | $\eta_1$ | $\eta_2$ | $g_{max}$ |
|-----------|----------|----------|----------|-----------|
| 300 | 10 | 0.08 | 0.008 | 100 |
| $\alpha$ | $\beta$ | $\chi$ | $\mu$ | $k$ |
| 0.5 | 0.005 | 0.005 | 0.5 | 1000 |

The scenario of this experiment is as follows. While the sensor is moving straight ahead over flat terrain, we suddenly put an obstacle in front of it. The result of the experiment is depicted in Figure 9, with an associated video link in Supplementary Video S8.

We compare our improved topological generator with the current topological map generator. The difference is summarized in Figure 9c: our improved generator can clarify the obstacle by increasing the density of topological structure, while the other generator could not. In our improved dynamic attention processor, the topological structure associated with safe, flat terrains is tenuous. On detecting an obstacle, map density in the obstacle area promptly increases. Thus, the shape of the obstacle can be clarified. Such attention-focusing happens also in the human attentional system. When humans walk on safe terrain, only a low level of attention is paid toward the surrounding environment. However, on noticing an obstacle, attention increases, targeting the obstacle. Our improved model could hence contribute to the realization of effective locomotion by passing its attentional findings on to inform higher-level control.

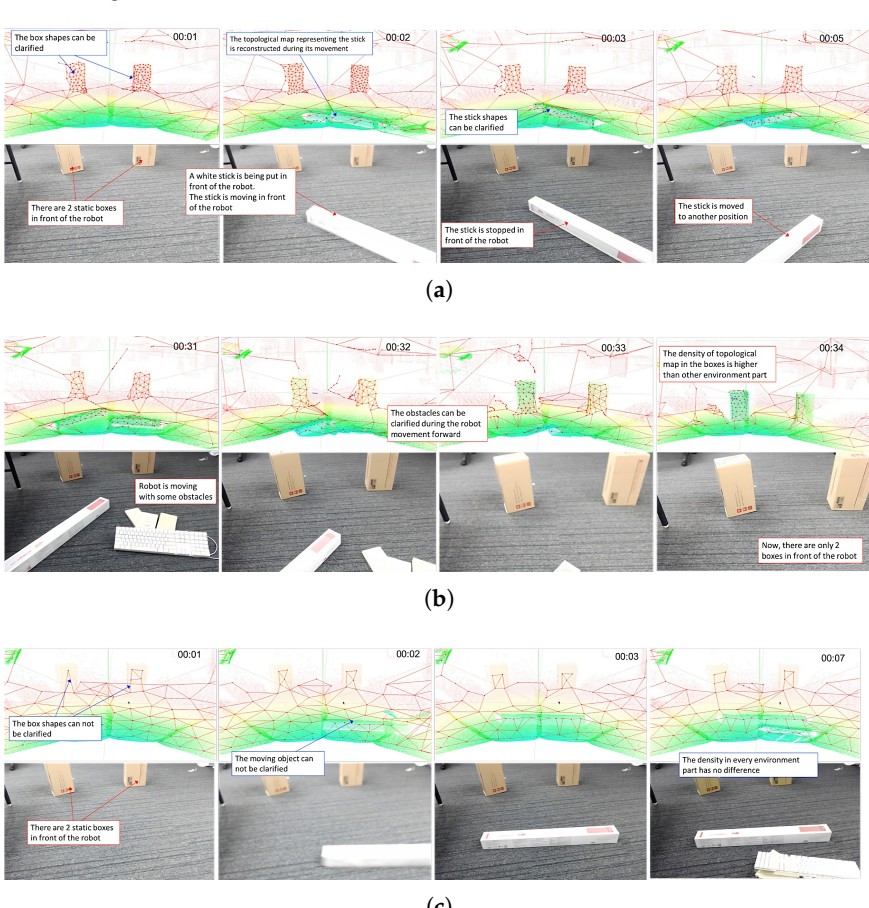

**Figure 9.** Result of dynamic attention model represented by topological map structure using point cloud data as the input. Video footage can be seen in Video S8. (**a**) An obstacle is suddenly put in front of the robot (**b**) The robot is moving through the obstacles (**c**) The result of the current topological map generator using another growing neural gas when an obstacle is put in front of the robot.

### 6.2. Experiment on Attention–Affordance in Locomotion

In Section 6.1, we showed how the dynamic attention mechanism is represented in the topological map while the robot is moving. The attention system is consistently aware of, or in other words is focused on, any obstacles blocking its way. Here, we will show the relationship between attention and affordance in robot locomotion from an ecological psychology viewpoint.

Affordances are always attached to objects whether they are perceived or not. However, affordances should be at least detected before being perceived, and they may be perceived accurately or inaccurately. When people walking around in a shopping mall, their attention is typically focused on the stores around them. Affordances, however, are still perceived with at least minimum accuracy.

To demonstrate how affordance and attention are integrated in robot locomotion, we first examine the attention and affordance results in simulation shown in Figure 10. The topological structure and the green ball represent the Attention model and the next foothold position of the current swinging leg, respectively. In the simulation, we put a sudden obstacle 0.1 s after the leg starts swinging in the area of next foothold position. When 0.1 s after obstacle was given, the non-homogeneous nodes were appearing. In this time, the affordance detects an obstacle with low accuracy, although the obstacle does not obstruct the foothold placement. The affordance sends the command to the attention model to increase the density of the node in the suspected area. Then, 0.11 s after, the number of nodes increased and the affordance was able to perceive the obstacle with high accuracy. In this time, the affordance asked to move the swinging model to the safety area. This experiment proves that the attention may affect the accuracy of detection of affordance and cause the failure decision.

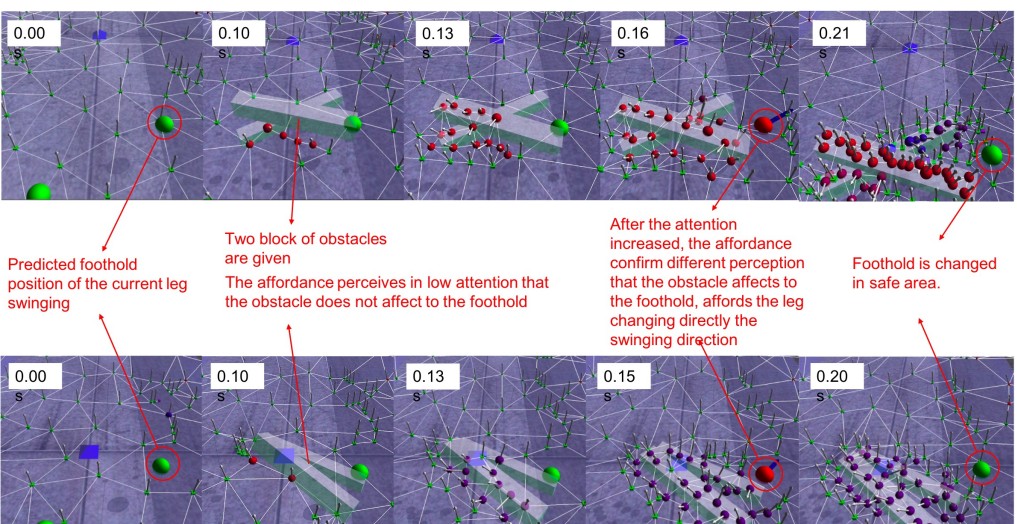

**Figure 10.** Mechanism of Integration between proposed Attention module and the Affordance model using computer simulation ODE. The video can be seen at Supplementary Video S7.

Furthermore, this experiment shows that the attention processor effectively contributes to affordance perception. In high-attention areas, the object's clarification implies that higher-level processes can have affordance information with higher accuracy. From the two experiments in Figure 10, the affordance detector perceived that the obstacle did not affect the foothold position's viability when attention to that area was low. After the attention there increases, the affordance detector perceives the region differently, and directs the robot to change the swing of its limb.

#### Attention–Affordance in Real Robot Performance

The result of perceived affordance with high accuracy and low accuracy, tested in a real robot, can be seen in Figure 11. This accuracy is influenced by the attention condition.

Figure 11a, Case 1 illustrates the affordance results of robot attention. The robot may be able to move in the green area. When obstacles are suspected in the robot's path, the attention is stimulated (see Case 2 in Figure 11b). Some nodes of the GNG roughly represent the obstacle's shape, and stimulate greater attention for clearer information about the obstacle. In the figure, the system seeks to clarify details of the obstacle indicated by a red-colored box.

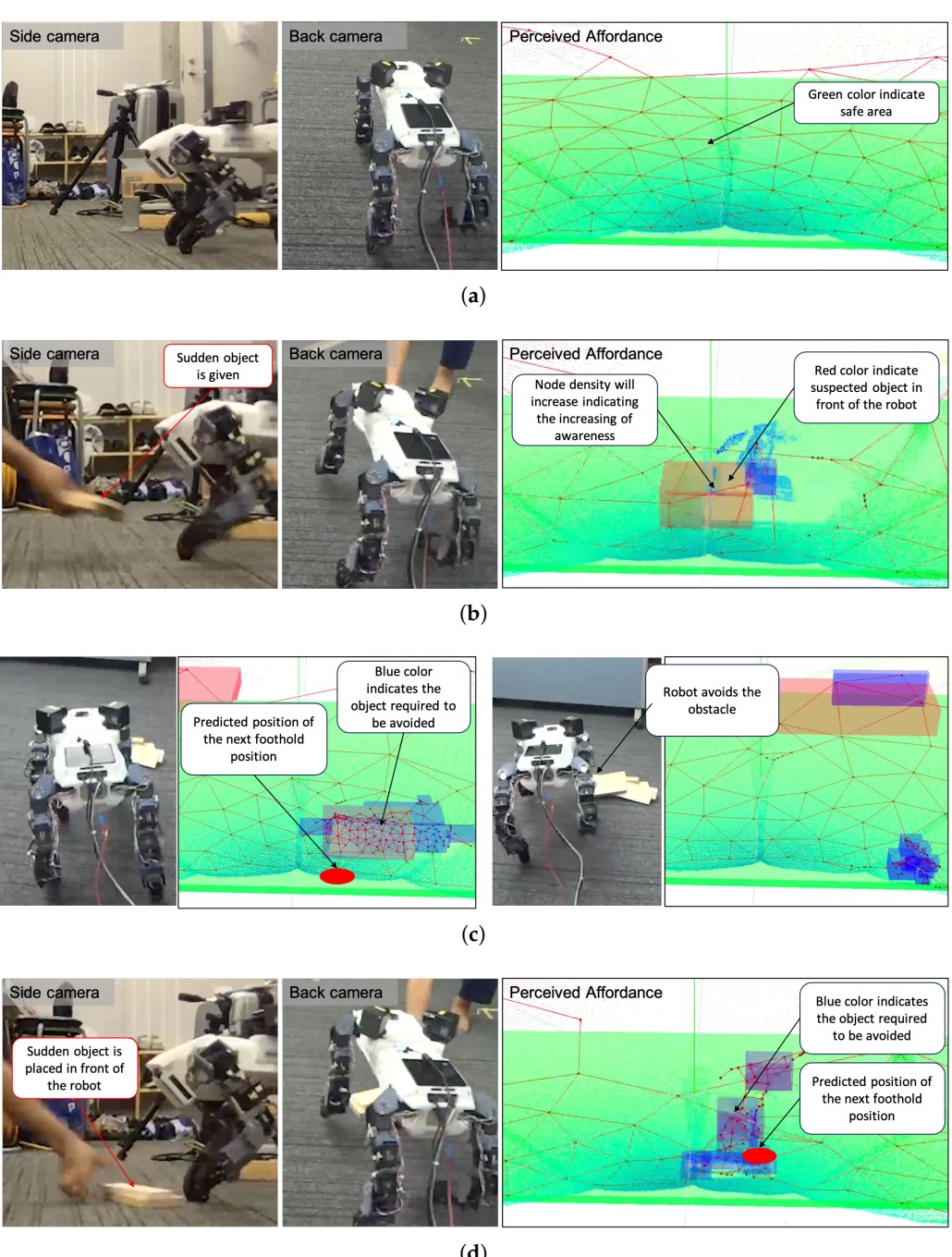

**Figure 11.** The result of different condition perceptions in different levels of data (3D point-cloud being the raw data from the depth sensors, topological map structure representing the attention model, perceived affordance, and generated action) (**a**) Case 1, there is no interference (**b**) Case 2, a sudden obstacle is given (**c**) Case 3, a sudden obstacle is placed in front of the robot with far distance (**d**) Case 4, a sudden obstacle is placed in front of the robot with short distance.

Once the density is fully increased and the object is sufficiently clarified, the affordance detector asks the robot to avoid the object. Since the object does not interfere with the swing currently under way, the robot decides to change its walking direction (see Figure 11c Case 3). Where the object position is in the way of the currently swinging limb, the predicted outcome is obviously collision with the obstacle (see Figure 11d Case 4). In this case, the movement generator changes the current swinging pattern.

Through this scenario, our attention model has demonstrated effective use of affordance detection in robot locomotion, using an environmental psychology model for integration between individual and environment [48,49]. Affordances may be wrongly perceived when attention is low, and perceived more accurately when attention is increased.

### 6.3. Experiment of Locomotion Pattern Generator

In order to develop an appropriate locomotion pattern generator, we optimized the interconnection between neurons in the central pattern generator (**w**) and its interconnection with motor neurons in motor neuron pools **x** (see Figure 6). We applied a multi-objective evolutionary algorithm for the optimization [58]. The fitness calculation optimizes for the desired speed and the energy efficiency. In this experiment, we want to optimize the robot to move at 0.5 m/s with minimal energy expenditure. The parameter setting for the locomotion model and NSGA is similar to the previous research [39], shown in Table 3.

**Table 3.** Locomotion Optimization Parameters.

| Parameter | Value |
|---|---|
| $\tau$; $T$; $\tau_f$ | 12.0; 1.2; 1.0 |
| $b$; $u_0$ | 2.5; 1.0 |
| Time cycle | 0.01 s |
| Population size | 128 indv. |
| Number of generations | 100 gen., 100 gen. |
| Number of objectives | 2 obj. Speed and Stabilization: |
| Chromosome | 7 real numbers |
| Crossover & Mutation prob. | 0.3, 0.3 |
| Time evaluation | 10 s |

There is one parameter representing the combined neuron structure, 64 parameters representing the weight values between neurons in the CPG structure, and 12 parameters representing the connections between sensor and CPG neurons to be optimized. The parameter interconnection map is illustrated in Figure 12. For example, the weight parameters from CPG neuron 1 to CPG neuron 2 and from CPG neuron 2 to CPG neuron 1 are represented by parameter $C_1$. Then the weight parameter from CPG neuron 3 to CPG neuron 1 is represented by parameter $C_{10}$. Red values are the combination of the leg's neuron structure. This combination has a similar strategy with [59]. Blue values are the weight parameters between neurons in the CPG structure. Green values are the weight parameters between the touch sensors and the CPG's neurons.

| | | RECEIVER | | | | | | | | | | | | | | | | | | | | | | | |
|---|---|---|---|---|---|---|---|---|---|---|---|---|---|---|---|---|---|---|---|---|---|---|---|---|---|
| | | LEG 1 | | | | | | LEG-A | | | | | | LEG-B | | | | | | LEG-C | | | | | |
| | | 1 | 2 | 3 | 4 | 5 | 6 | 1 | 2 | 3 | 4 | 5 | 6 | 1 | 2 | 3 | 4 | 5 | 6 | 1 | 2 | 3 | 4 | 5 | 6 |
| **TRANSMITTER** / LEG 1 | 1 | - | $C_1$ | $C_2$ | $C_3$ | $C_4$ | $C_5$ | $C_{53}$ | $C_{54}$ | $C_{55}$ | $C_{56}$ | - | - | $C_{41}$ | $C_{42}$ | $C_{43}$ | $C_{44}$ | - | - | $C_{29}$ | $C_{30}$ | $C_{31}$ | $C_{32}$ | - | - |
| | 2 | $C_1$ | - | $C_6$ | $C_7$ | $C_8$ | $C_9$ | $C_{57}$ | $C_{58}$ | $C_{59}$ | $C_{60}$ | - | - | $C_{45}$ | $C_{46}$ | $C_{47}$ | $C_{48}$ | - | - | $C_{33}$ | $C_{34}$ | $C_{35}$ | $C_{36}$ | - | - |
| | 3 | $C_{10}$ | $C_{11}$ | - | $C_{12}$ | $C_{13}$ | $C_{14}$ | $C_{61}$ | $C_{62}$ | - | - | - | - | $C_{49}$ | $C_{50}$ | - | - | - | - | $C_{37}$ | $C_{38}$ | - | - | - | - |
| | 4 | $C_{15}$ | $C_{16}$ | $C_{12}$ | - | $C_{17}$ | $C_{18}$ | $C_{63}$ | $C_{64}$ | - | - | - | - | $C_{51}$ | $C_{52}$ | - | - | - | - | $C_{39}$ | $C_{40}$ | - | - | - | - |
| | 5 | $C_{19}$ | $C_{20}$ | $C_{21}$ | $C_{22}$ | - | $C_{23}$ | - | - | - | - | - | - | - | - | - | - | - | - | - | - | - | - | - | - |
| | 6 | $C_{24}$ | $C_{25}$ | $C_{26}$ | $C_{27}$ | $C_{23}$ | - | - | - | - | - | - | - | - | - | - | - | - | - | - | - | - | - | - | - |
| LEG 2 | 1 | $C_{29}$ | $C_{30}$ | $C_{31}$ | $C_{32}$ | - | - | - | $C_1$ | $C_2$ | $C_3$ | $C_4$ | $C_5$ | $C_{53}$ | $C_{54}$ | $C_{55}$ | $C_{56}$ | - | - | $C_{41}$ | $C_{42}$ | $C_{43}$ | $C_{44}$ | - | - |
| | 2 | $C_{33}$ | $C_{34}$ | $C_{35}$ | $C_{36}$ | - | - | $C_1$ | - | $C_6$ | $C_7$ | $C_8$ | $C_9$ | $C_{57}$ | $C_{58}$ | $C_{59}$ | $C_{60}$ | - | - | $C_{45}$ | $C_{46}$ | $C_{47}$ | $C_{48}$ | - | - |
| | 3 | $C_{37}$ | $C_{38}$ | - | - | - | - | $C_{10}$ | $C_{11}$ | - | $C_{12}$ | $C_{13}$ | $C_{14}$ | $C_{61}$ | $C_{62}$ | - | - | - | - | $C_{49}$ | $C_{50}$ | - | - | - | - |
| | 4 | $C_{39}$ | $C_{40}$ | - | - | - | - | $C_{15}$ | $C_{16}$ | $C_{12}$ | - | $C_{17}$ | $C_{18}$ | $C_{63}$ | $C_{64}$ | - | - | - | - | $C_{51}$ | $C_{52}$ | - | - | - | - |
| | 5 | - | - | - | - | - | - | $C_{19}$ | $C_{20}$ | $C_{21}$ | $C_{22}$ | - | $C_{23}$ | - | - | - | - | - | - | - | - | - | - | - | - |
| | 6 | - | - | - | - | - | - | $C_{24}$ | $C_{25}$ | $C_{26}$ | $C_{27}$ | $C_{23}$ | - | - | - | - | - | - | - | - | - | - | - | - | - |
| LEG 3 | 1 | $C_{41}$ | $C_{42}$ | $C_{43}$ | $C_{44}$ | - | - | $C_{29}$ | $C_{30}$ | $C_{31}$ | $C_{32}$ | - | - | - | $C_1$ | $C_2$ | $C_3$ | $C_4$ | $C_5$ | $C_{53}$ | $C_{54}$ | $C_{55}$ | $C_{56}$ | - | - |
| | 2 | $C_{45}$ | $C_{46}$ | $C_{47}$ | $C_{48}$ | - | - | $C_{33}$ | $C_{34}$ | $C_{35}$ | $C_{36}$ | - | - | $C_1$ | - | $C_6$ | $C_7$ | $C_8$ | $C_9$ | $C_{57}$ | $C_{58}$ | $C_{59}$ | $C_{60}$ | - | - |
| | 3 | $C_{49}$ | $C_{50}$ | - | - | - | - | $C_{37}$ | $C_{38}$ | - | - | - | - | $C_{10}$ | $C_{11}$ | - | $C_{12}$ | $C_{13}$ | $C_{14}$ | $C_{61}$ | $C_{62}$ | - | - | - | - |
| | 4 | $C_{51}$ | $C_{52}$ | - | - | - | - | $C_{39}$ | $C_{40}$ | - | - | - | - | $C_{15}$ | $C_{16}$ | $C_{12}$ | - | $C_{17}$ | $C_{18}$ | $C_{63}$ | $C_{64}$ | - | - | - | - |
| | 5 | - | - | - | - | - | - | - | - | - | - | - | - | $C_{19}$ | $C_{20}$ | $C_{21}$ | $C_{22}$ | - | $C_{23}$ | - | - | - | - | - | - |
| | 6 | - | - | - | - | - | - | - | - | - | - | - | - | $C_{24}$ | $C_{25}$ | $C_{26}$ | $C_{27}$ | $C_{23}$ | - | - | - | - | - | - | - |
| LEG 4 | 1 | $C_{53}$ | $C_{54}$ | $C_{55}$ | $C_{56}$ | - | - | $C_{41}$ | $C_{42}$ | $C_{43}$ | $C_{44}$ | - | - | $C_{29}$ | $C_{30}$ | $C_{31}$ | $C_{32}$ | - | - | - | $C_1$ | $C_2$ | $C_3$ | $C_4$ | $C_5$ |
| | 2 | $C_{57}$ | $C_{58}$ | $C_{59}$ | $C_{60}$ | - | - | $C_{45}$ | $C_{46}$ | $C_{47}$ | $C_{48}$ | - | - | $C_{33}$ | $C_{34}$ | $C_{35}$ | $C_{36}$ | - | - | $C_1$ | - | $C_6$ | $C_7$ | $C_8$ | $C_9$ |
| | 3 | $C_{61}$ | $C_{62}$ | - | - | - | - | $C_{49}$ | $C_{50}$ | - | - | - | - | $C_{37}$ | $C_{38}$ | - | - | - | - | $C_{10}$ | $C_{11}$ | - | $C_{12}$ | $C_{13}$ | $C_{14}$ |
| | 4 | $C_{63}$ | $C_{64}$ | - | - | - | - | $C_{51}$ | $C_{52}$ | - | - | - | - | $C_{39}$ | $C_{40}$ | - | - | - | - | $C_{15}$ | $C_{16}$ | $C_{12}$ | - | $C_{17}$ | $C_{18}$ |
| | 5 | - | - | - | - | - | - | - | - | - | - | - | - | - | - | - | - | - | - | $C_{19}$ | $C_{20}$ | $C_{21}$ | $C_{22}$ | - | $C_{23}$ |
| | 6 | - | - | - | - | - | - | - | - | - | - | - | - | - | - | - | - | - | - | $C_{24}$ | $C_{25}$ | $C_{26}$ | $C_{27}$ | $C_{23}$ | - |
| Sensor | 1 | $S_1$ | $S_2$ | $S_3$ | $S_4$ | $S_5$ | $S_6$ | $S_{11}$ | $S_{12}$ | - | - | - | - | $S_9$ | $S_{10}$ | - | - | - | - | $S_7$ | $S_8$ | - | - | - | - |
| | 2 | $S_7$ | $S_8$ | - | - | - | - | $S_1$ | $S_2$ | $S_3$ | $S_4$ | $S_5$ | $S_6$ | $S_{11}$ | $S_{12}$ | - | - | - | - | $S_9$ | $S_{10}$ | - | - | - | - |
| | 3 | $S_9$ | $S_{10}$ | - | - | - | - | $S_7$ | $S_8$ | - | - | - | - | $S_1$ | $S_2$ | $S_3$ | $S_4$ | $S_5$ | $S_6$ | $S_{11}$ | $S_{12}$ | - | - | - | - |
| | 4 | $S_{11}$ | $S_{12}$ | - | - | - | - | $S_9$ | $S_{10}$ | - | - | - | - | $S_7$ | $S_8$ | - | - | - | - | $S_1$ | $S_2$ | $S_3$ | $S_4$ | $S_5$ | $S_6$ |

**Figure 12.** CPG and Internal Sensory Interconnection.

After achieving 100 generations, the robot can move with the desired speed and with the minimum energy required. The fitness evolution of all solutions in every generation is depicted in Figure 13. The optimized signal of CPG neurons transmitted to MNs network before being generated to the joint angle level is shown in Figure 14.

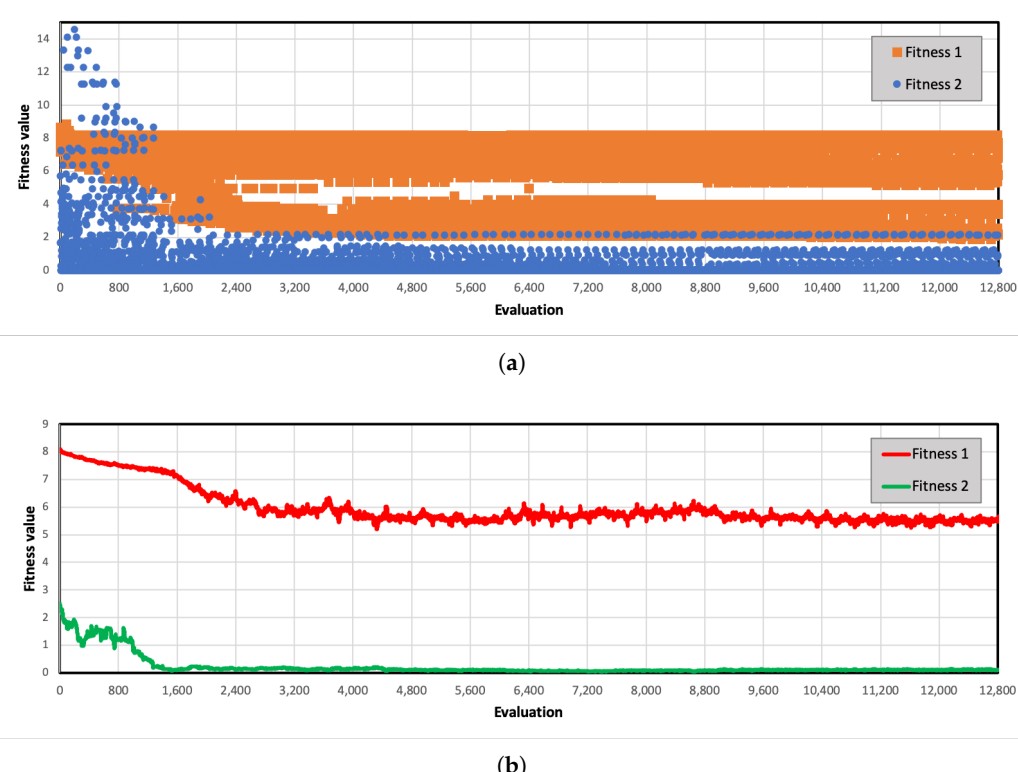

(**a**)

(**b**)

**Figure 13.** The evolution of fitness over all generations. Orange points represent the first fitness value, and blue points represent the second fitness value. (**a**) Fitness values of all solutions (**b**) Fitness values averaged over all solutions.

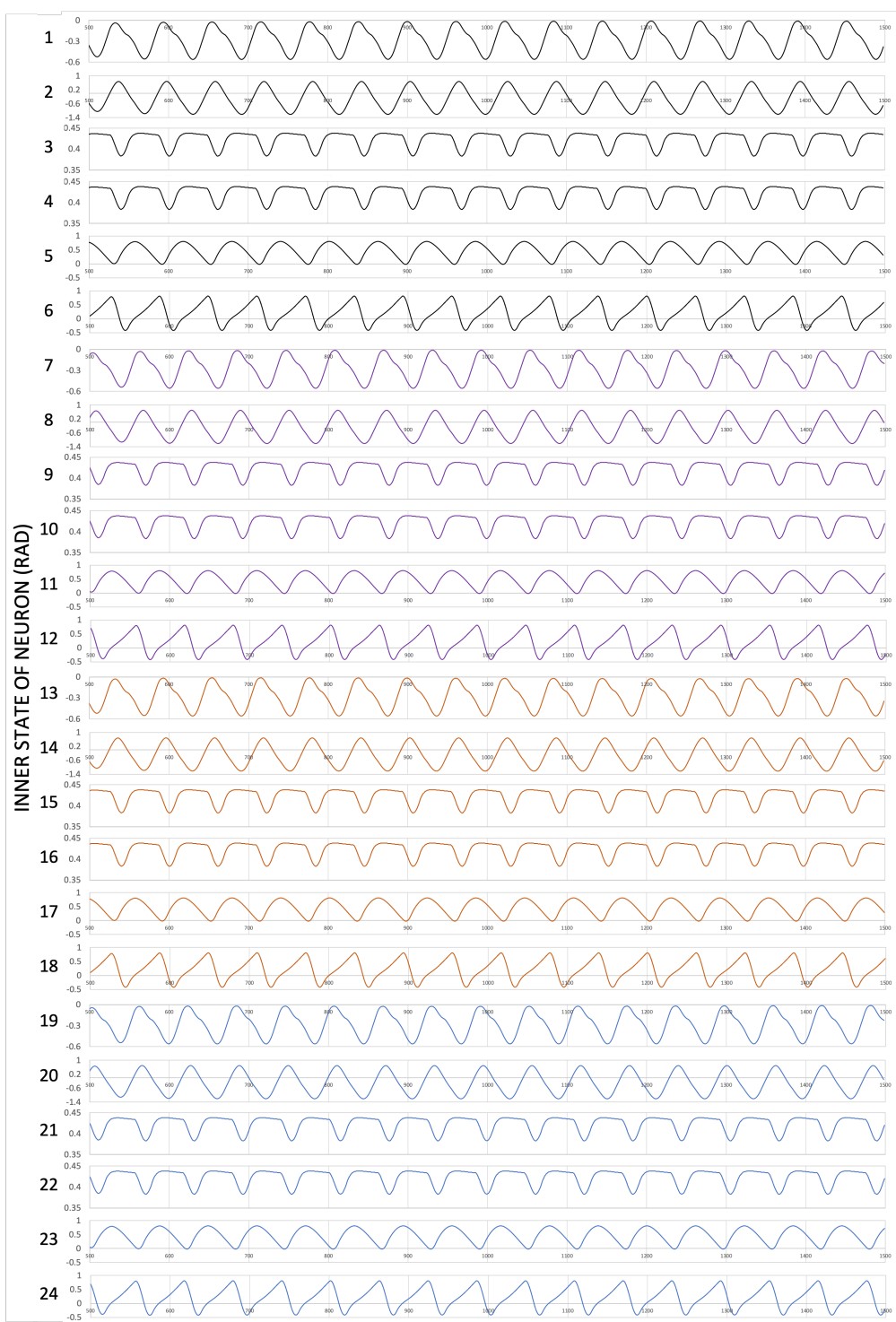

**Figure 14.** Output of inner state of every CPG neuron.

### 6.4. Experiment of Affordance–Effectivity Fit

This experiment aims to optimize appropriate action based on the information perceived from the affordance detector and the body posture condition (torso tilt angle, leg joint angles). We optimized the model, represented by ANN, in ODE and MATLAB. First, we record the movement of the leg's end effector to acquire the reference data. We record the current vector speed of the leg's end effector ($\mathbf{v}(t)$), the previous joint angle ($\theta(t-1)$), the angular acceleration ($\ddot{\theta}(t-1)$), and the touching condition ($\mathbf{T}(t-1)$). The result of the recorded random leg movement can be seen in Figure 15. After that, we optimize the ANN

structure of AEF in MATLAB using backpropagation method, based on the affordance information, recorded movement references, and the robot's body posture. Once optimized, we conduct the robot performance trial in computer simulation and in a real robot.

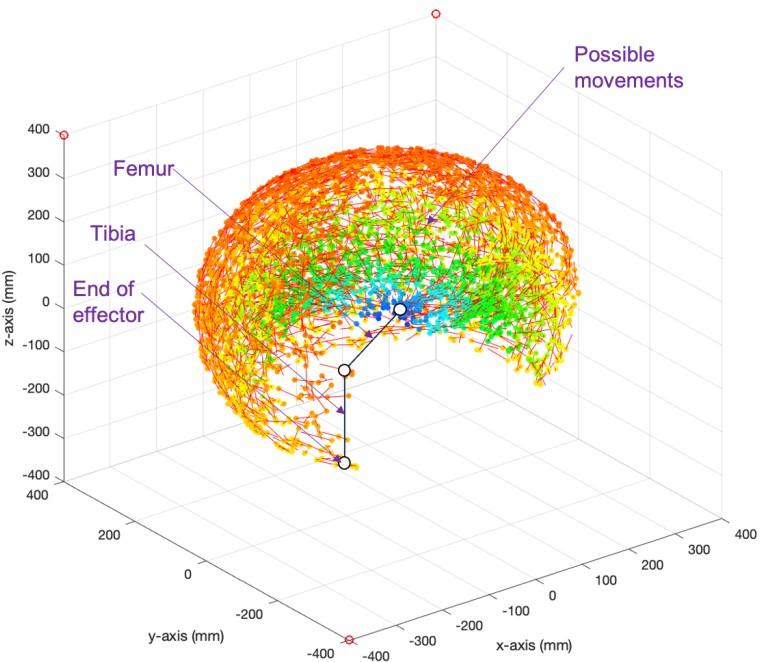

**Figure 15.** The recorded movement vectors of leg's end effector as reference data generated from random movement of the leg. The color represents the distance of movement from the base of the leg. Circle dot represents the initial position of the effector end, and the line represents the vector direction of movement.

Implementation in Computer Simulation

In this experiment, the robot moved in a straight direction with the optimized locomotion model explained in Section 5. On the way, there are three small blocks as obstacles. The heights of the blocks are 6, 7, and 8 cm. As shown in Figure 16a, the robot can avoid the obstacle by changing the leg swing pattern and/or changing the footing position. Based on the information given by affordance detection model, when the interrupt is activated as depicted in Figure 17, then the motion interrupt generator sends the interrupt signal at the joint angle level. As shown in Figure 18, the step height changes due to the influence of the motion interrupt generator. The sample of pattern signals from CPG and joint angle signals is shown in Figure 19. The signal pattern is not influenced. Only the signal from the motor neuron pools is influenced. This mechanism can realize good performance as well. However, further discussion and development are required. When we disabled our improved process, the robot got stuck in the obstacle. Snapshots of this failure are shown in Figure 16b. Detailed video can be seen in Supplementary Video S1.

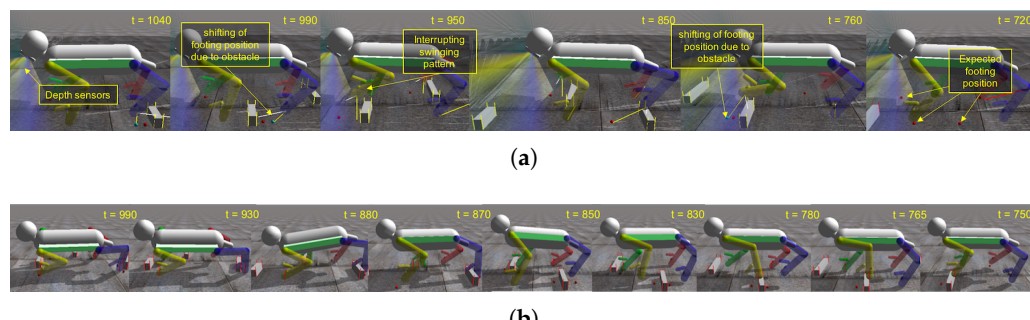

**(a)**

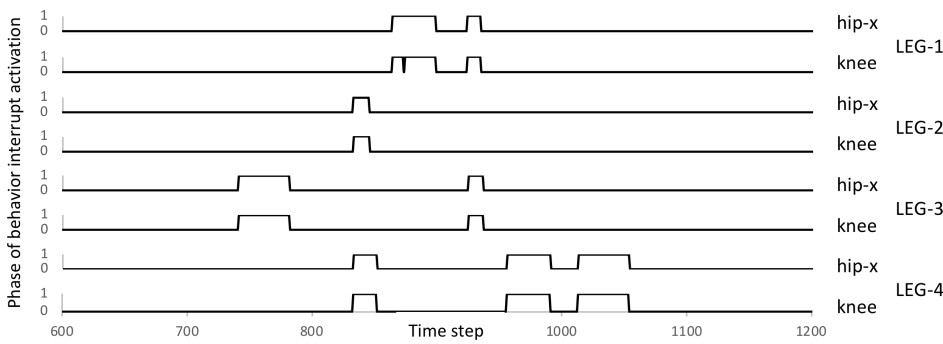

**(b)**

**Figure 16.** Snapshots of simulation performance. Detailed video can be seen in Supplementary Video S1 (**a**) the improved process enabled (**b**) without the improved process.

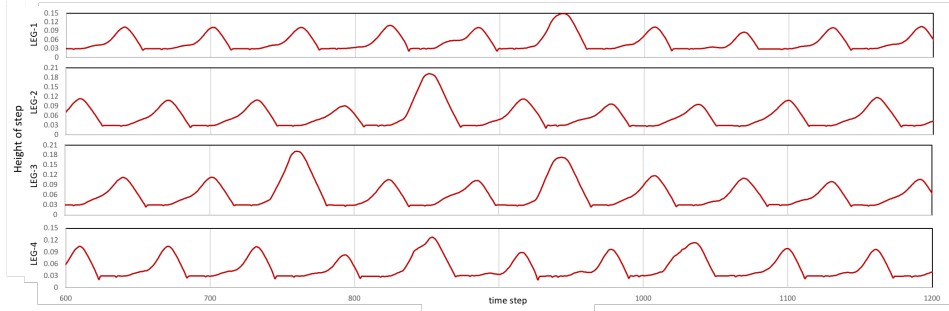

**Figure 17.** Recorded interrupted swinging phase during the performance.

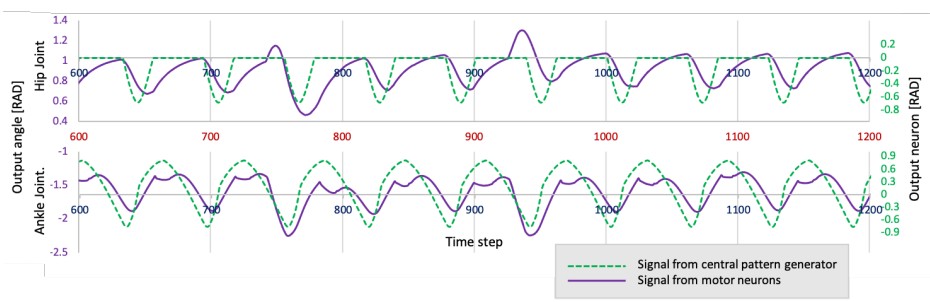

**Figure 18.** Recorded height of feet in every time cycle.

We performed additional experiments showing the performance of the proposed module in different situations. These can be seen in the following Supplementary Video S2. The green circle indicates that the next robot step is in the safe area. The red circle indicates that the next robot step is in the unsafe area. The robot will step the green area of topological nodes. Furthermore, in this simulation, we also show the performance of robot movement on the moving stepping stone. Detailed video can be seen in Supplementary Video S3.

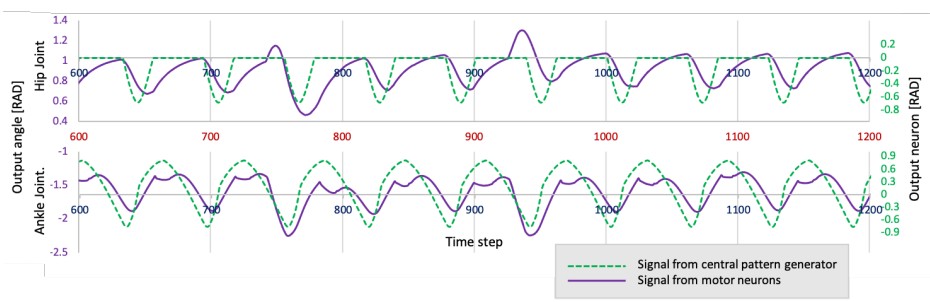

**Figure 19.** The output of the central pattern generator in green line and the output from motor neurons in purple line.

### 6.5. Quantitative Experiment of Whole Proposed Model

Furthermore, in order to show the effectiveness of the overall adaptive control method, we conducted some quantitative experiments: (1) avoiding a sudden obstacle, and (2) avoiding a static obstacle. We evaluate the response of obstacle detection and the accuracy of avoiding. Evaluating the accuracy of avoiding, we classify the outcome into three categories: obstacle totally avoided, obstacle avoided but touched, and avoidance failed.

#### 6.5.1. Robot Performance Avoiding Sudden Obstacle

We run the quadruped robot in a trotting gait at normal speed. During the swinging phase (front leg lifted), some pieces of wood are moved and placed by hand immediately in front of the robot. We repeated this activity 50 times with random positions and arrangements of the obstacle. The affordance detection process successfully clarified all of the obstacles presented this way. The performance results can be seen in Table 4. In 28 of the trials, the robot totally avoided the obstacle, in 13 trials the robot avoided but touched the obstacle, and nine times, the robot failed to avoid it. The success rate for avoiding the obstacle thus reached 82 percent. Snapshots of the robot avoiding the obstacle can be seen in Figure 20a. A link to the associated video can be seen at Supplementary Video S4 and the quantitative experiments can be seen at Supplementary Video S5.

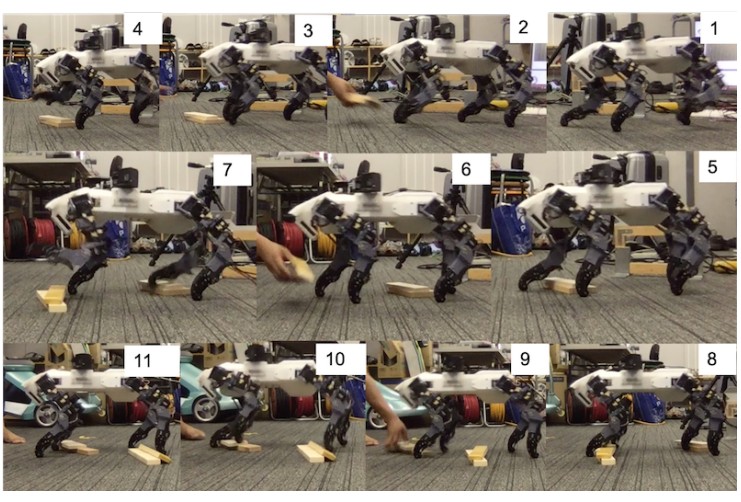

(**a**)

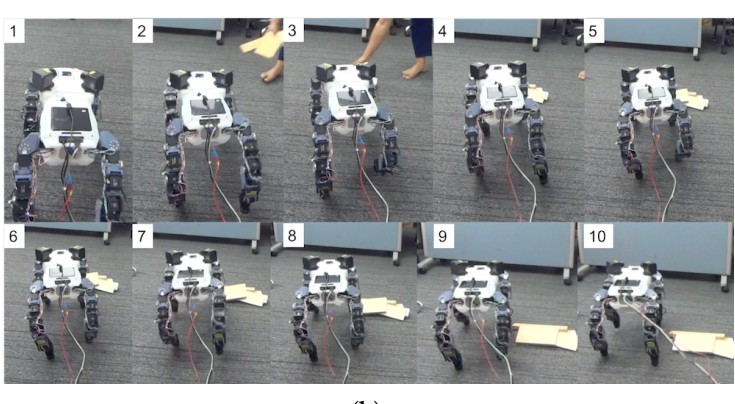

(**b**)

**Figure 20.** Result of real robot performance success in avoiding obstacle using the proposed model. The video can be seen at Supplementary Video S4 and the quantitative experiments can be seen at Supplementary Video S5. (**a**) The robot changed the swinging pattern of its front leg, responding to the sudden obstacle in the swinging area. (**b**) The robot moved left to avoid an obstacle further ahead.

**Table 4.** The Result of Quantitative Experiments.

| Quantitative Experiments | Success with Touching | Success Total | Failed |
|---|---|---|---|
| Robot performance avoiding sudden obstacle | 13 | 28 | 9 |
| Robot performance avoiding static obstacle | 21 | 23 | 6 |

6.5.2. Robot Performance Avoiding Static Obstacle

We set the quadruped robot moving with a normal-speed walking gait. We put some obstacle across the robot's path fifty times with random obstacle arrangements. In this experiment, the affordance detector also succeeded in clarifying the obstacle. The performance results can be seen in Table 4. The robot successfully avoided the obstacle 23 times, avoided but touched the obstacle 21 times, and failed six times. Overall, the robot's success rate was 88 percent. Snapshots of a sample performance can be seen in Figure 20b.

6.5.3. Surface Matching for Preciseness Analysis of Attention Model

This process compares the location and vector curvature of nodes between the Attention and the Reference structure. There are three steps to take: (1) Locate the nearest Reference topological node to compare the Attention node, (2) Perform 6D vector-matching similarity, and (3) Calculate rate of similarity.

First, the module locates the nearest reference's node to relocate the selected Attention node. It locates the map node with the closest location and normal vector to each Attention node. If the module cannot locate it inside the specified range, the node will be skipped to the next phase. Only nodes assigned to the map are chosen to be processed in the following stage. The illustration of the surface matching can be seen in Figure 21. The reference node's location and surface vector difference are computed using Equations (12) and (13). Parameter $\mathbb{A}$ and $\mathbb{Q}$ represent attention nodes and reference node, respectively. The distance difference between $d_i$th attention node and $j$th reference's node is calculated as follows:

$$l_{d_i,j} = \|\mathbb{Q}_j^n - \mathbb{A}_{d_i}^n\| \tag{12}$$

Then, the degree of vector difference between $d_i$th attention node and $j$th reference's node is calculated as follows

$$v_{d_i,j} = \hat{\mathbb{Q}}_j^v + \hat{\mathbb{A}}_{d_i}^v \tag{13}$$

The module selects the closest node in a small region with a high similarity value for the surface vector.

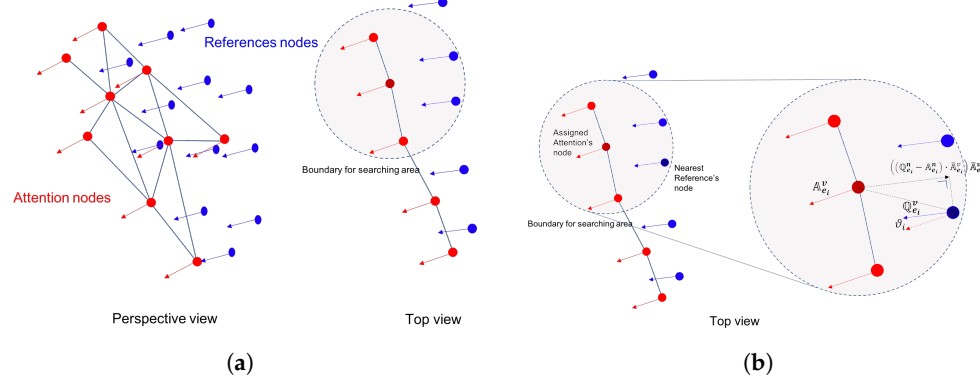

**Figure 21.** Illustration of the surface matching (**a**) Different between attention node and reference node (**b**) Calculation of surface maching

Second, the assigned Attention nodes are matched to the reference node. It is required to calculate the difference of 3D position and 3D Euler angle. These are calculated as follows:

$$\dot{s}(t) = \frac{1}{N_d} \sum_{i=1}^{N_d} \left( (\mathbb{Q}_{e_i}^n - \mathbb{A}_{d_i}^n) \dot{\mathbb{A}}_{d_i}^v \right) \hat{\mathbb{A}}_{d_i}^v \tag{14}$$

$$\dot{\theta}(t) = \frac{1}{N_d} \sum_{i=1}^{N_d} (\mathbb{Q}_j^\theta - \mathbb{A}_{d_i}^\theta) \tag{15}$$

where $\dot{s}(t)$ is the difference in position.

Third, based on the difference value of node position and 3D Euler angle, the normalized similarity rate is calculated as follows:

$$\mu = 0.5 \frac{e_{max}^p - min(\dot{s}, e_{max}^p)}{e_{max}^p} + 0.5 \frac{e_{max}^t - min(\dot{s}, e_{max}^t)}{e_{max}^t} \tag{16}$$

where $e_{max}^p$ and $e_{max}^t$ are the maximum error of position and 3D Euler angle.

6.5.4. Failure Analysis

To analyze the failures, we compare the similarity rate between attention topological structure and reference structure of obstacle. The comparison between success case and failure case can be seen in Table 5.

**Table 5.** Failure Analysis

| Condition | Success | Failed |
|---|---|---|
| Maximum rate of similarity achieved (average) | 0.92 | 0.89 |
| Time achieving maximum rate (average) | 0.45 | 0.75 |
| Average similarity achieved in 0.5 s | 0.92 | 0.48 |

The best similarity rate between success case and failure case is almost the same, which is about 0.9. However, in the timing achieving best or saturated similarity rate, the success case, which is 0.45 s, is faster than the failure case, which is 0.75 s. The failure case only achieved a similarity rate of 0.5 in 0.45 s. The process of attention module affected the generated action. If the attention process has a low similarity rate, it may cause failure in the positioning of the swinging leg.

## 7. Advantage and Comparison with Existing Model

Based on the experimental results, the advantages of the improved system can be treated in three categories: (1) Development of attention model using topological structure, (2) Integration between attention and affordance in moving behavior, (3) Integration of exteroceptive sensory information to lower-level control of locomotion generator in short-term adaptation. However, the new system's robustness should be improved towards achieving an obstacle avoidance success rate over 90 percent. Furthermore, currently only the limbs are active to respond to a sudden obstacle. In the future, we will also address the torso's posture behavior.

### 7.1. Development of Attention Model Using Topological Structure

Attention can be represented by any parameter, depending on the data processed. In our case, we use only time-of-flight sensors for the external information; these generate 3D point-cloud data. We therefore chose the topological map model for optimal data representation. However, the existing topological map building process offers no way to control node density in only a localized area. Our contribution addresses that through a dynamic density topological map-building process based on a growing neural gas. We

put a strength parameter into every node to indicate the importance of each node in the topological structure. This parameter will be defined based on the segmentation process in the previous time. The experiment in Section 6.1 shows that the improved dynamic attention process can clarify obstacles in front of the robot by increasing the number of nodes.

Other topological generator systems such as Self-Organizing Map (SOM), Growing Cell Structure (GCS), and Neural Gas (NG), cannot increase node density in localized areas. Therefore, they need to increase the node density over the entire map in order to clarify even localized objects [60,61]. Compared with other multi-density topological maps, such as multi-layer growing neural gas (ML-GNG) [20], our improved system may decrease the processing time by as much as 70 percent (ML-GNG = $3.1567 \times 10^{-4}$ $s$, DD-GNG = $1.0255 \times 10^{-4}$ ). The localized attention-focusing process has also been proven to decrease the computational cost.

### 7.2. Building the Integration between Attention and Affordance

When affordance and attention are strongly integrated, affordance can be perceived with high accuracy. From the experimental results, the attention controller provides the topological structure information to the affordance detector. Then, the affordance detector will interpret the environmental information by calculating the surface normal vectors. When the normal vectors indicate a suspected obstacle, the attention controller will increase the node density, and hence the information density, in the suspected area. We can reduce computational overhead by reducing the raw data points. The attention controller also automatically increases the node density where needed to clarify suspected objects. This mechanism is efficient for cognitive processing since only important information is processed. In contrast with existing methods [62,63], our affordance detection can be 10 times faster in comparison with the 1.36 ms required by [63].

### 7.3. Integration of Exteroceptive Sensory Information to Lower-Level Control of Locomotion Generator in Short-Term Adaptation

From the experiment conducted, we can prove that our improved system integrates low-level control and exteroceptive sensory information and of the locomotion generator. As a result, the system can adapt to the changing of movement area at any time step. The current locomotion models that combine perceptions and locomotion mostly focus on foothold planning, which restricts adaptability to the sudden changing of foothold surface area [16,24,63–67]. While [68,69] proposed legged locomotion that is able to avoid some obstacles in real time, their algorithm cannot handle impediments that obstruct a swinging limb in the middle of it. Their success rate was 76%. However, they have different experimental procedures. Our model has the advantage of having the capability to control the locomotion in a lower time period. This has been demonstrated in various trials involving the avoidance of unknown unexpected obstacles.

## 8. Discussion and Conclusions

We proposed a new concept in locomotion generation that integrates the cognitive model from an ecological psychology viewpoint. Our concept decreases the gap between the cognitive model and the locomotion model, and changes overall system structure from hierarchical to parallel. This concept is based on the perceiving–acting cycle in ecological psychology. In this paper, we emphasize the attention controller as the starting process for handling exteroceptive sensory information required by further cognitive processes. We also provide the video demonstration in Supplementary Video S6 to show the essence of the proposed model.

We develop a novel attention controller for movement behavior using a topological structure. It processes 3D point-cloud data as exteroceptive sensory information. We can control the density of topological structures in the localized areas to clarify details of objects in the robot's surroundings. To realize the proposed attention model, we build

a dynamic density topological model based on GNG, called DD-GNG. Compared with other topological reconstruction, our improved system may decrease the processing time by as much as 70 percent (ML-GNG = $3.1567 \times 10^{-4}$ $s$, DD-GNG model = $1.0255 \times 10^{-4}$ ). The proposed model also has a higher rate similarity of obstacles comparing with the stated model. The proposed model reaches 0.91 and the other stated model reaches 0.6 due to the topological densities.

In order to integrate the relationship between attention and locomotion, we must consider affordances. The experiment in Section 6.1 shows that the attention will affect the accuracy of affordance detection, and the affordance detector influences the required action. Our proposed affordance detector provides semantic function in locomotion behavior. It can provide object information in the context of the robot's current capabilities. We succeeded in contributing to the implementation of integration of attention and affordance in robot locomotion.

In order to prove the effectiveness of the proposed system, we conducted robot locomotion trials in both simulation and in real robot performance. In the simulation, when the robot encounters a sudden obstacle, the robot is able to avoid the obstacle by changing its limb swinging pattern. In the real experiment, the robot is able to avoid the sudden obstacle by changing its swinging pattern and its foothold target. This experiment proves that our proposed model integrates the cognitive and locomotion generator in short-term adaptation, in a way that can be understood through concepts from ecological psychology. Furthermore, when the obstacle is a little further ahead of the robot, the robot in both simulation and real performance changed its direction of movement to walk around the obstacle. The proposed system is hence effective enough to warrant further development for different applications and more advanced experiments. The proposed system is not limited to quadrupedal robots; it can be applied to robots with any number of legs, or any other kind of mobility.

Overall, based on validation by the experiments, the contributions of this paper are as follows:

- We proposed a dynamic-density topological map construction process that can generate different densities in a localized area for detected objects;
- We proposed real integration between attention and affordance in robot locomotion based on ecological psychology;
- We developed a novel locomotion system that integrates external sensory information in short-term adaptation.

There are still many possibilities for implementing the dynamic attention model, especially in mobile robot applications. In our research plan, we will continue to develop the locomotion generator part that considers more sensory feedback. Our concept of neuro-cognitive locomotion has high prospects of achieving dynamic locomotion that integrates cognition with the locomotion generator.

**Supplementary Materials:** A supporting video article is available at link. Supplementary Video S1: https://www.dropbox.com/s/jh4q26gvvkjktzi/video%201.mp4?dl=0. Supplementary Video S2: https://www.dropbox.com/s/6o7o0cz5j39up95/video%202.mp4?dl=0. Supplementary Video S3: https://www.dropbox.com/s/8bwwcuh7jymdbvd/video%203.mp4?dl=0. Supplementary Video S4: https://www.dropbox.com/s/t6yasv7a4zdzpwv/video%204.mp4?dl=0. Supplementary Video S5: https://www.dropbox.com/s/tzx2z23j85bkd92/video%205.mp4?dl=0. Supplementary Video S6: https://www.dropbox.com/s/zuk23u10xyldy8f/Overall%20video.mp4?dl=0. Supplementary Video S7: https://www.dropbox.com/s/hlftolpnkckdafl/video%207.mp4?dl=0. Supplementary Video S8: https://www.dropbox.com/s/h0d0iobqeq1nfdt/video%208.mp4?dl=0.

**Author Contributions:** A.A.S. contributed to the design and development of the robot's hardware. A.A.S. and J.B. contributed to building the learning and optimization model. A.A.S. and N.K. contributed to building the neural-cognitive based model. All authors contributed to writing and refining the manuscript. All authors have read and agreed to the published version of the manuscript.

**Funding:** This work was partially supported by JST [Moonshot RnD][Grant Number JP- MJMS2034].

**Institutional Review Board Statement:** Not applicable.

**Informed Consent Statement:** Not applicable.

**Data Availability Statement:** Not applicable.

**Conflicts of Interest:** The authors declare no conflict of interest.

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
