# Peer review of "Neuro-Cognitive Locomotion with Dynamic Attention on Topological Structure"

_machines, doi:10.3390/machines11060619_

Round 1
Reviewer 1 Report
This paper presented a neuro-cognitive locomotion model for locomotion robots including a dynamic attention model, a movement-related affordance detection model and a locomotion generator model. The proposed model was also evaluated by simulations and real quadruped robots. Generally, the presented concept is interesting. However, several important issues should also be addressed to improve the quality of this manuscript. Below are some comments for the authors to consider:
1. The presentation of Figure 3 can be improved. There should be some descriptions in the caption to illustrate subfigure (a) and (b). In the current presentation, the authors have put three photos in Figure 3(b) to show the sensor, while the three photos provided the similar information. Please indicate the function of the second and third photo.
2. In line 225 and 232, the authors have used two algorithms from reference [43]. However, it is also necessary to provide more details of these two algorithms directly in this manuscript rather than referring to the reference, because they are important for the readers to understand the proposed model.
3. In Figure 6, the presented diagram is exactly the same as in Figure 2. So what is the purpose of Figure 6?
4. In Figure 7 of page 11, the authors only provided a caption without figure. Please clarify.
5. Please provide some reasons or illustrations for choosing the parameters in Table 2 (page 13).
6. In Figure 7 of page 14, the sequence of the subfigures [c, d, e] was inconsistent with that in the caption [a, b, c]. Please clarify.
7. In Figure 10 of page 17, please provide the x-axis-label for subfigure (b).
8. The text in Figure 11 is too small to read. Please improve that.
9. In Figure 14, please indicate the meaning of the red and blue curves in the figure caption.
10. Please illustrate the meaning of the symbol Q and A in Equation (12) and (13).
11. In the current state of the art, there are many studies using soft legs to realize complex movements of locomotion robots or quadruped robots. The authors are also recommended to include those work in the Introduction or Discussion section of this manuscript and discuss the feasibility of the proposed model for the soft-leg-based locomotion robots. Below are several recommended references:
Tolley, M. T. et al. (2014). A resilient, untethered soft robot. Soft Robotics 1(3), 213–223. DOI: https://doi.org/10.1089/soro.2014.0008.
Sun, Y., Zong, C., Pancheri, F., Chen, T., & Lueth, T. C. (2023). Design of topology optimized compliant legs for bio-inspired quadruped robots. Scientific Reports, 13(1), 4875. DOI: https://doi.org/10.1038/s41598-023-32106-5.
Li, Y., Fish, F., Chen, Y., Ren, T., & Zhou, J. (2019). Bio-inspired robotic dog paddling: kinematic and hydro-dynamic analysis. Bioinspiration & Biomimetics, 14(6), 066008. DOI: https://doi.org/10.1088/1748-3190/ab3d05.
Reviewer 2 Report
This paper proposed a new concept in locomotion generation that integrates the cognitive model from an ecological psychology viewpoint, and developed a novel attention controller for movement behavior using a topological structure. The topic of this paper is closely related to the hot topics in the field of mobile robots, and content is substantial. The theory, simulation and experiment are complete. To further improve the quality of the paper, the following suggestions are put forward:
1. The abstract of the paper mainly explains how to integrate attention and perception to produce behavior. If the advantages and innovation of the method in the paper are further explained, the paper may be more attractive;
2. There are a few formatting problems in the paper. The format of the reference is not standard; Too much content in some pictures leads to small text size, it is suggested to integrate or remove some less important content, or split the big picture into multiple smaller pictures. The screenshots in FIG. 9a and FIG. 12 are incomplete, so I suggest export pictures instead of screenshots. The area of data points in FIG. 9 is too large to affect the change rule of the curveï¼›
3. What is the basis for setting the parameters in Table 2 and what are the advantages of setting the parameters in this way. Explaining it might make the paper more convincing;
4. The scenes in this paper are mainly flat terrain with sufficient background light and large color contrast between obstacles and background. If the scenes are closer to reality, does the method in this paper have adaptability to it.
Minor editing of English language required
Reviewer 3 Report
The article deals with problems associated with quadruped locomotion. This area is extremely topical because the development of legged robots has recently been the subject of research by many scientific teams. Legged locomotion is biologically inspired locomotion, which is an inspiring model for us on how to implement locomotion. It is necessary to solve many technical problems, such as solving the kinematics and dynamics of the robot's movement, the stability of the robot, generating optimal movements for the creation of legged locomotion. Furthermore, it is necessary to solve obstacle avoiding, intelligent trajectory planning, energy-optimized planning of the implementation of locomotion and many other problems.
This article discusses the mechanism of integration of locomotion with cognition in robots. The proposed concept evaluates external and internal sensory information and identifies optimal robot activities with regard to the robot's current locomotion capabilities. The use of legged locomotion was clumsy, demanding and energetically disadvantageous, but by adding algorithms for learning, cognitive control of movements approaches the locomotion of animals that realize their locomotion in this way.
In the introduction, the authors present the need and importance of solving this issue and present the current state in this area.
The authors' goal is to realize a more human-like behavior by integrating neuro-cognitive functions into robot locomotion. The robot has to use both interoceptive sensory information about its own body and also exteroceptive sensory information about its surroundings. Integrating these sensory domains can be difficult when cognition is implemented separately from the locomotion generator.
The authors will try to find answers to the following scientific questions:
scientific questions:
• How to build a locomotion model that integrates both exteroceptive and interoceptive sensory information?
• How can exteroceptive information effect lower-level locomotion control (short-term adaptation)?
• How can we build an attention model for robot locomotion using point-cloud data?
A neuro-cognitive locomotion model that integrates strongly with both exteroceptive and interoceptive sensory information was designed for this purpose.
Diagram of neuro-cognitive locomotion and overall system design to locomotion generator structure to affordance effectiveness fit model are presented in this work.
The experiments confirmed the effectiveness of the use of individual modules in this work. Experiments with obstacles, where it was necessary to implement obstacle avoidance, are interesting and important.
contributions of this paper are summarized as:
• Authors proposed a dynamic-density topological map construction process that can generate different densities in localized areas, for detected objects.
• They proposed the real integration between attention and affordance in robot locomotion based on ecological psychology.
• Also they developed a novel locomotion system that integrates external sensory information in short-term adaptation.
I consider the achievement of these benefits and innovations in this work to be extremely important also from the point of view of the possibility of implementation into real locomotion systems in practice.
The article is prepared at an excellent level and the technical presentation is excellent. The research results are clearly presented. I did not find any serious flaws and errors in the article.
Even so, I have a few comments to improve the article.
Comments:
I miss the presented kinematic model of the used locomotion system in the article. I am asking for a graphic presentation of the kinematic model.
The basic parameters of the used experimental model are also missing.
In Figure 10, it is cut off from the description of the x-axis.
Round 2
Reviewer 1 Report
The authors have revised the manuscript according to my comments. Therefore, I recommend to publish this paper in this journal.